# Discovering Differences in Strategic Behavior between Humans and LLMs

**Caroline Wang** [1]   **Daniel Kasenberg** [2]   **Kim Stachenfeld** [2]   **Pablo Samuel Castro** [2]

## Abstract

As Large Language Models (LLMs) are increasingly deployed in social and strategic scenarios, it becomes critical to understand where and why their behavior diverges from that of humans. While behavioral game theory (BGT) provides a framework for analyzing behavior, existing models do not fully capture the idiosyncratic behavior of humans or black-box, non-human agents like LLMs. We employ AlphaEvolve, a cutting-edge program discovery tool, to directly discover *interpretable* models of human and LLM behavior from data, thereby enabling open-ended discovery of structural factors driving human and LLM behavior. Our analysis on iterated rock-paper-scissors reveals that frontier LLMs can be capable of deeper strategic behavior than humans. These results provide a foundation for understanding structural differences driving differences in human and LLM behavior in strategic interactions.

## 1. Introduction

Artificial intelligence (AI) is becoming ubiquitous in human activities, fundamentally altering how we work, communicate, and interact. In particular, the rapid advancement of Large Language Models (LLMs) has catalyzed a shift toward agents capable of generating human-passing text in interactive social settings. These agents are frequently anthropomorphized by users and researchers alike, although they are intrinsically non-human.

We identify two predominant scenarios: first, where LLM agents interact *directly* with humans and other artificial agents, and second, where LLM agents are used as *proxies* to study human behavior. Examples of 'direct interaction' scenarios are numerous, as they often pose direct economic value, and include general-purpose conversational assistants (Ouyang et al., 2022), coding assistants (Srinivasan et al., 2025), customer service agents (Buhler, 2025), virtual relationship agents (Siemon et al., 2022), and negotiation agents (Van Hoek & Lacity, 2023). Simultaneously, driven by the discovery that groups of LLMs can display interesting and human-like emergent social behavior (Park et al., 2022), researchers in the social sciences and social computing, as well as market research firms (Korst et al., 2025; Amble & Cohen, 2025), have begun to use LLM agents as a convenient and cheap method to simulate human behavior in both individual and social scenarios (Ziems et al., 2024).

As users continue to interact with LLMs in their everyday lives, and researchers publish findings using LLM proxies for human behavior, the following question becomes critical for both safety and scientific progress: *What are the similarities and differences between LLM and human behavior in repeated social interactions?* Understanding differences is crucial for understanding limitations of LLMs as digital twins, improving behavioral alignment to human expectations, and monitoring LLM capabilities. While answering this question represents a broad and ambitious research endeavor, this paper studies the question through the lens of the iterated matrix game, rock-paper-scissors. Games have long been used to measure cognitive abilities for both humans (Allen et al., 2024) and AI (Mnih et al., 2015; Silver et al., 2016). Matrix games, in particular, exemplify real-world social and strategic scenarios, thereby allowing us to study strategic interactions in a controlled sandbox.

Our approach draws from established methods in behavioral game theory (BGT; Camerer, 2003). BGT seeks to understand *human* social behavior through empirically studying behavior in simple games. There are two dominant methods of inquiry: (1) studying simple behavioral statistics (e.g., the win rate in rock-paper-scissors), and (2) constructing parameterized mathematical models to fit behavioral data, and studying those models (Zhu et al., 2025; Wright & Leyton-Brown, 2019). While behavioral statistics are essential for identifying trends in data, they do not explain them. On the other hand, traditional BGT models rely on manually designed mathematical formulas as *hypotheses* to explain and predict deviations in human behavior from

[1]Department of Computer Science, University of Texas at Austin. Work performed as a student researcher at Google DeepMind [2]Google DeepMind. Correspondence to: Caroline Wang <caroline.l.wang@utexas.edu>.

*Proceedings of the 43rd International Conference on Machine Learning*, Seoul, South Korea. PMLR 306, 2026. Copyright 2026 by the author(s).

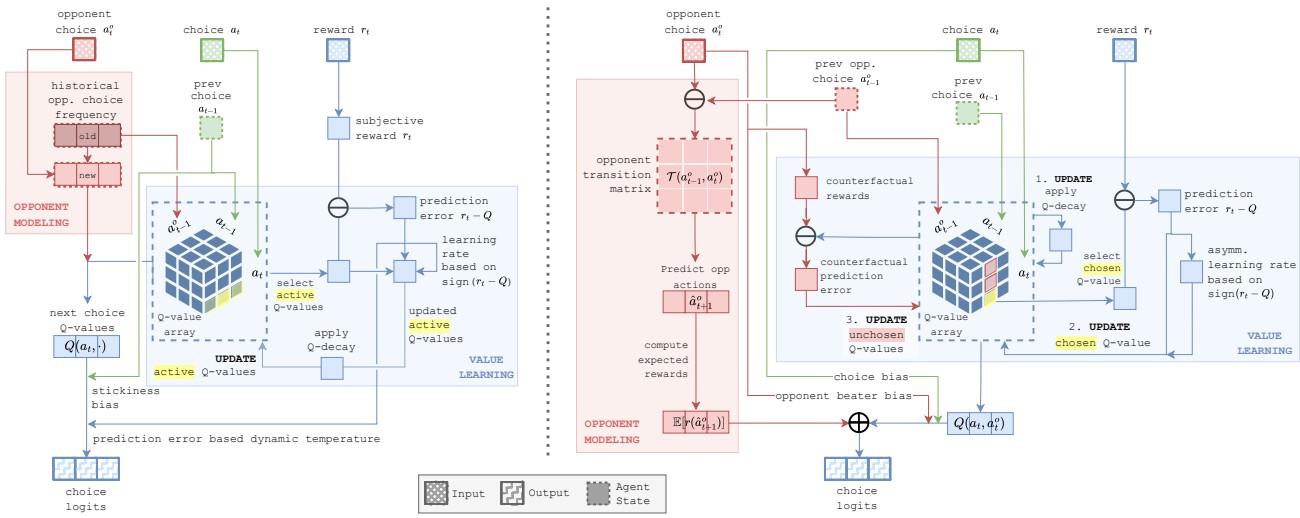

*Figure 1.* **Diagrams of AlphaEvolve-discovered programs.** The simplest-but-best programs for humans (*left*) and Gemini 2.5 Pro (*right*), discovered by AlphaEvolve on iterated rock-paper-scissors, are displayed. For simplicity, the learnable parameters $\theta$ of each program are omitted. Both programs use value-based learning and opponent modeling. The Gemini 2.5 Pro program displays more sophisticated opponent modeling than humans and considers counterfactual outcomes during value updates.

game theoretically rational behavior.[1] However, as LLMs introduce non-human behavioral priors, relying solely on human-centric hypotheses may fail to capture facets of LLM strategic behavior that differ from humans.

In this work, we aim to structurally characterize the differences between human and LLM behavior in strategic scenarios without limiting our framework to models designed for human behavior. We automate the model generation process by using LLMs to generate interpretable models of behavior that fit the data well (Novikov et al., 2025; Castro et al., 2025; Rmus et al., 2025; Aygün et al., 2025). This allows us to explore the space of candidate models supported by behavioral data from humans and various LLMs, to understand behavioral differences on a structural level. We compare the behavior of humans and LLMs in the matrix game Iterated Rock-Paper-Scissors (IRPS), a simple but foundational testbed that is actively used to study human strategic behavior (Batzilis et al., 2019), cognitive biases (Dyson et al., 2016), neural processes (Moerel et al., 2025), opponent modeling (Brockbank & Vul, 2024), and evolutionary dynamics (Wang et al., 2014).

Our work differs from existing analyses of LLM strategic behavior because existing analyses focus on LLM strategic capabilities in isolation (Cipolina-Kun et al., 2025; Payne & Alloui-Cros, 2025; Akata et al., 2025), or analyze game-specific statistics (Roberts et al., 2025; Fan et al., 2024). In contrast, we analyze a data-driven space of models generated by LLMs, which allows for open-ended discovery of

factors driving both human and LLM behavior.

Our contributions are as follows:

- We introduce the first application of automated symbolic model discovery to characterize human and LLM behavior through interpretable behavioral models.
- We demonstrate that frontier LLMs win with higher rates and against more complex opponents than humans in IRPS, even without agentic scaffolds.
- Using AlphaEvolve-discovered behavioral models, we provide a structural explanation for this performance gap, revealing that frontier LLMs maintain more sophisticated opponent models than humans.

**Conflict of Interest Disclosure**   All authors were employed by Google over the course of this study. Google leads the development of Gemini models, which are among the ones evaluated in this paper.

## 2. IRPS as a Testbed

Rock-paper-scissors (RPS) is a two-player zero-sum game with well-understood theoretical properties. The Nash equilibrium, which specifies game-theoretically optimal behavior, prescribes that a rational agent should act randomly to prevent exploitable patterns in behavior from developing (von Neumann et al., 1944). However, humans do not behave randomly in RPS (Batzilis et al., 2019; Wang et al., 2014; Xu et al., 2013), motivating a large body of work across BGT, neuroscience, and cognitive psychology, despite the game's simplicity. Studies have demonstrated that humans exhibit "primary salience bias", favoring Rock on the first move (Xu et al., 2013; Brockbank & Vul, 2024),

---

[1]Behavioral game theorists have also considered neural network models, which may fit the data arbitrarily well, but this shifts the onus from model design to model interpretation (Hartford et al., 2016).

and employ conditional response strategies (e.g., win-stay-lose-shift; Dyson et al., 2016).

Of particular interest is that *iterated* rock-paper-scissors (IRPS) creates the ideal conditions to investigate the human ability to perform *opponent modeling*—a form of theory-of-mind (ToM; Premack & Woodruff, 1978). Due to the game's simplicity, the only way to consistently win is to rapidly detect strategic regularities in the opponent's behavior and adapt accordingly, which humans do to a limited extent (Brockbank & Vul, 2024; Batzilis et al., 2019). Consequently, IRPS has been used as a benchmark for both AI game-playing (Lanctot et al., 2023) and ToM in LLMs (Cross et al., 2025).

ToM, or the cognitive ability for an agent to understand the beliefs, desires and intentions of others, is a crucial aspect of human social interaction. Thus, we identify IRPS as a particularly interesting domain in which to compare human and LLM strategic capabilities. We leverage an existing dataset of human gameplay on IRPS (Brockbank & Vul, 2024), and construct a *matched* dataset for each LLM to enable a rigorous comparison. The IRPS setting and each dataset is described below. Further details about each dataset are provided in App. D.

**IRPS Game and Bots**   The IRPS game setting is inherited from Brockbank & Vul (2024). Each game lasts for 300 rounds. In each round, players receive 3, 0, or -1 for a win, tie, or loss respectively.[2] Opponents consist of 15 bots of varying complexity; see App. C for details). For each game, the opponent is one of 15 bots of varying complexity.

The bots can be divided into nonadaptive and adaptive bots. *Nonadaptive* bots employ transition-based strategies that map a small amount of historical context to fixed next moves (e.g., playing the move that beats the opponent's prior move) with 90% probability. *Adaptive* bots track historical sequential dependencies in their opponent's moves, and play to counter the predicted next move with 100% probability. Importantly, all bots are fully exploitable by an opponent that tracks the bot's patterns in behavior.

**Human Dataset**   Brockbank & Vul (2024) introduced a large-scale dataset where 411 human participants play IRPS against bots for a total of 129,087 choices. Participants were informed that the bots' strategies were fixed, but not what each bot strategy was. After each round, participants were shown both moves and the round outcome. Throughout the game, participants were also shown the game instructions, and a tally indicating progress through the game.

[2]Values were chosen by Brockbank & Vul (2024) to maintain engagement by increasing opportunity to gain points. Values do not change the Nash equilibrium.

|          | rock     | paper    | scissors |
|----------|----------|----------|----------|
| rock     | (0, 0)   | (-1, 3)  | (3, -1)  |
| paper    | (3, -1)  | (0, 0)   | (-1, 3)  |
| scissors | (-1, 3)  | (3, -1)  | (0, 0)   |

*Table 1.* Rewards for one round of the IRPS game. Each tuple contains the rewards of the (row, column) players.

**Matched LLM Datasets**   To enable a fair comparison to behavior in the existing human dataset, data is collected under conditions that match those of the Brockbank & Vul (2024) dataset as closely as possible. In particular, the LLMs play IRPS using the same game payoffs described previously, where gameplay prompts are based on the instructions given to the humans. For each LLM, 20 games of 300 rounds were collected against all 15 bots, for a total of 90,000 choices.

We consider reasoning-capable, closed and open-source LLMs from Google and OpenAI, that were the most advanced at the time of this study: Gemini 2.5 Pro and 2.5 Flash (Comanici et al., 2025), GPT 5.1 (OpenAI, 2025b), and the open-source GPT OSS 120B (OpenAI, 2025a). Data collection details are provided in App. D.2.

## 3. Methods

We distill the underlying behavior in each dataset via *AlphaEvolve* (Novikov et al., 2025), a recently introduced framework for generating programs that maximize a mathematical objective. By using LLMs to evolve Python programs that describe behavioral data, AlphaEvolve bridges the gap between high-performance but black-box models, and interpretable, theory-driven equations. While discovered programs are optimized for a predictive loss and may not correspond to actual causal mechanisms of behavior, nevertheless, they provide human-readable, mechanistic descriptions that can be analyzed and verified like traditional scientific hypotheses. Consequently, they offer both competitive predictive accuracy and mechanistic interpretability. Below, we provide the problem formulation for behavioral modeling in IRPS, followed by an example of a programmatic behavioral model. We then describe the baseline models as well as how AlphaEvolve is used to discover programmatic behavioral models directly from data.

### 3.1. Problem Formulation: Behavior Modeling in IRPS

We consider the problem of predicting an agent's choices at each round of a two-player game, given its prior actions, the opponent's prior actions, and prior game outcomes. The IRPS game can be represented using the formalism of the two-player, symmetric, iterated game, $\Gamma = \langle A, r, T \rangle$. In the game $\Gamma$, $A = A^1 = A^2$ represents the finite set of actions

available to each player, $r : A^1 \times A^2 \mapsto \mathbb{R}^2$ specifies the outcome of each round by associating numeric rewards to each pair of actions, and $T$ represents the number of rounds. For IRPS, $A$ is the set {rock, paper, scissors}, $r$ is specified in Table 1, and $T = 300$. The subscript $t$ will denote the game round, and $a, a^o$ will respectively refer to the actions of the *ego agent*, whose perspective we take, and its opponent.

A *behavioral model* is a function $\phi(a_t, a_t^o, r_t, h_t, \theta) \rightarrow (\hat{p}_{t+1}, h_{t+1})$ that processes the current ego agent action, $a_t$, the current opponent action, $a_t^o \in A$, and the received reward $r_t$, to update an internal state $h_t$ and output a probability distribution over the ego agent's possible next moves $\hat{p}_{t+1} \in \Delta(A)$. A behavioral model is parameterized by a vector $\theta$, which may be *fit* to a dataset $\mathcal{D} = \{\tau_i\}_{i=1}^N$, using the standard maximum likelihood estimation and stochastic gradient descent (SGD). In the definition of $\mathcal{D}$, $N$ denotes the number of games, and $\tau$, the sequence of choices and rewards within each game, $\tau = \{a_t, a_t^o, r_t\}_{t=1}^T$. See App. E.1 for further details. In the following, we will use $\phi(\theta_\mathcal{D})$ to indicate what dataset $\phi$ is fit to.

A *programmatic behavioral model* is a model that is represented as a program (App. Fig. 4). In the literature, behavioral models have consisted of human-specified mathematical update rules based on behavioral mechanisms, or machine learning models, such as neural networks (Camerer & Hua Ho, 1999; Camerer et al., 2004; Hartford et al., 2016; Zhu et al., 2025). Both types are expressible as programs, underscoring that programs provide highly flexible representation for behavioral models. Our baselines include both types, while AlphaEvolve programs fall into the latter category.

Within the context of this study, we outline two criteria for ideal behavioral models. First, models should predict behavior well, even on unseen data (*generalization*), to ensure that our conclusions are not simply due to overfitting (Hartford et al., 2016). Second, they should be *interpretable*, to allow us to gain insight on differences between behavioral models of humans and LLMs. To evaluate predictive fit and generalization, model performance is reported as the two-fold cross-validated normalized likelihood (see App. E). To evaluate the interpretability of the programmatic behavioral models examined in this paper, we employ the Halstead effort, a software engineering heuristic designed to measure the time required to comprehend and implement a program, based on the number of total and unique operators and operands in code (detailed in App. E.5).

### 3.2. Evolving Behavioral Models using AlphaEvolve

We adopt the training procedure of Castro et al. (2025), who employed a predecessor to AlphaEvolve to generate programmatic cognitive models. The general framework

is briefly summarized below, focusing on distinctions of our framework from Castro et al. (2025). We refer the interested reader to App. E.4 for further details.

In brief, AlphaEvolve is an evolutionary optimization procedure that uses LLMs to generate programs that maximize a *fitness function*, which maps candidate programs to a scalar score. To generate a program, an LLM is provided with contextual information, a small sample of previously generated programs, a parent program, and corresponding scores of provided programs. The LLM is instructed to propose modifications to the parent program in order to increase its score. As LLMs have been trained on large amounts of data, they are capable of leveraging diverse, existing theories on human strategic behavior to propose candidate models.

We use AlphaEvolve to discover programmatic behavioral models that fit a dataset $\mathcal{D}$. There are two key distinctions between our approach and Castro et al. (2025). First, we use AlphaEvolve (Novikov et al., 2025), rather than its predecessor, FunSearch (Romera-Paredes et al., 2024). Second, to avoid overfitting and maximize program interpretability, we use a multi-objective fitness function that considers both the cross-validated likelihood of the training dataset, and the Halstead effort. This resembles the approach of Rmus et al. (2025), who used the Bayesian information criterion (BIC; Watanabe, 2013) to evaluate a programmatic model's predictive performance and simplicity.[3] The multi-objective optimization results in a Pareto frontier that trades off between predictive performance and interpretability (e.g., Figure 4).

```python
def agent(
    params: jnp.array,
    choice: int,
    opponent_choice: int,
    reward: float,
    agent_state: Optional[jnp.array],
) -> tuple[jnp.array, jnp.array]:
    """Behavioral model describing agent behavior on iterated rock-
        paper-scissors."""

    agent_state = None
    choice_logits = jnp.array([1.0, 1.0, 1.0])

    return choice_logits, agent_state
```

*Listing 1.* **Initial template program for AlphaEvolve.** All discovered programs follow this functional signature. The initial program is equivalent to the Nash equilibrium model.

In our AlphaEvolve configuration, Gemini 2.5 Flash is used to generate programs. AlphaEvolve is initialized with a template program that provides the desired program specifications, such as the input and output variables (Fig. 1). Note that the template program represents the parameters $\theta$ as the variable, params. Let $\Phi$ denote the space of programs, and

---

[3]While the BIC is a widely adopted approach for model selection in statistics, it is less suited for measuring the interpretability of a program. The BIC formula relies on the number of data points used to train a model and the number of parameters estimated by the model. In our setting, all programs generated by AlphaEvolve are trained on the same number of data points and have the same number of estimated parameters.

$\mathcal{F}(\phi) : \Phi \mapsto \mathbb{R}$ denote the fitness function over programs. In further detail, the first component of the fitness function is the twofold cross-validated maximum likelihood achieved by a program $\phi$, *after* its parameters $\theta$ are fit to dataset $\mathcal{D}$ via SGD (App. Eq. 2), i.e. $\mathcal{F}(\phi(\theta_\mathcal{D}))$. The overall AlphaEvolve procedure amounts to a bilevel optimization process, where the outer loop searches over the program space $\Phi$, while the inner loop searches over parameter space $\theta$. Thus, in contrast to standard machine learning methods that require the user to pre-specify the model structure (e.g., the architecture of a neural network), AlphaEvolve *automatically searches over possible model structures*.

### 3.3. Baseline Behavioral Models

To assess the accuracy of models discovered by AlphaEvolve, we compare them against three representative baselines. The first is the usual Nash equilibrium baseline, the second is founded on classic baselines in BGT, while the third uses highly flexible neural networks.

**Nash Equilibrium**   The classic game-theoretic baseline assumes all players act rationally. In IRPS, this corresponds to playing each move with probability $1/3$ (von Neumann et al., 1944).

**Contextual Sophisticated Experience-Weighted Attraction (CS-EWA)**   The majority of BGT models focus on non-repeated games, failing to capture within-game adaptation. While Experience-Weighted Attraction (EWA) (Camerer & Hua Ho, 1999) hybridizes reinforcement and belief-based learning for repeated play, and Sophisticated EWA (Camerer et al., 2002) incorporates recursive reasoning about opponents, both assume random rematching between rounds. This prevents modeling strategies that adapt to a specific opponent's history. To address this, we extend Sophisticated EWA to CS-EWA, which accounts for temporal dependencies by maintaining independent attraction vectors for every joint history of length $L = 2$. This allows the model to learn context-specific strategies (e.g., "if I played Rock and lost, switch to Paper"). See App. E.3 for details.

**Recurrent Neural Network (RNN)**   As a reference point for performance of a highly flexible but black-box statistical learner, we use an RNN based on the Gated Recurrent Unit (GRU; Cho et al., 2014). While highly flexible and capable of capturing complex sequential patterns in behavioral data, RNNs are challenging to interpret, and do not directly provide insight on behavior.

## 4. Comparing Human and LLM Strategic Behavior Using AlphaEvolve

We perform a structural comparison of human and LLM behavior by comparing the respective best behavioral models generated by AlphaEvolve. This section presents the

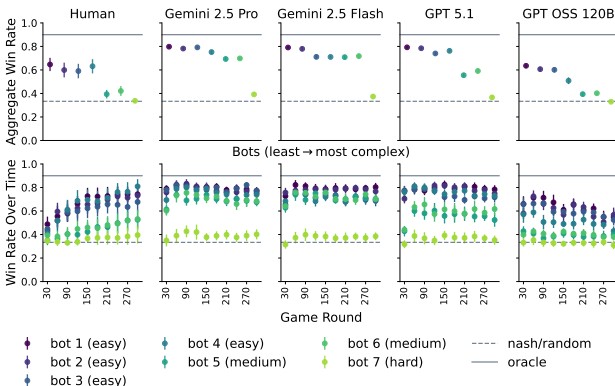

*Figure 2.* **Win rates of humans and LLMs against nonadaptive bots.** Bots are colored from least to most complex. The *top* row displays the aggregate win rate over all bots while the *bottom* row displays the win rate over time. Each column corresponds to a different agent type, with human win rates displayed in the leftmost column. Mean and 95% CIs are displayed, with some CIs not visible due to marker size.

experimental results, which are organized along three axes. First, to ground the comparison, we directly compare the win rate statistics of human and all LLMs, broken down by each bot type (Section 4.1), **finding that frontier LLMs outperform humans but follow similar aggregate trends.** Second, to validate the quality-of-fit for programs generated by AlphaEvolve, AlphaEvolve is compared to baseline behavioral modeling methods (Section 4.2), **demonstrating better behavior modeling performance than baselines.** Finally, we investigate structural differences between human and LLM behavior discovered by AlphaEvolve, by comparing the simplest-but-best generated programs (Section 4.3). We find that the **superior performance of strong LLMs stems from their capacity to maintain more complex opponent models.** Supplemental results and experimental details are presented in App. B.e

### 4.1. Win Rates of Humans and LLMs

The average win rates of each agent against each nonadaptive bot is reported in Fig. 2, compared to the win rates of a randomly acting player and a player with oracle knowledge of the bot policy. The analysis below focuses on the non-adaptive bots, while results for adaptive bots are provided in App. B.

We first consider the aggregate win rates (top row). Both humans and LLMs win at rates well above random chance against the simpler nonadaptive bots, with Gemini 2.5 Flash/Pro and GPT 5.1 winning at higher rates than humans, yet still beneath the optimal oracle win rate.[4] Win rates tend to decrease as the nonadaptive bots become more complex, with all agents failing to exploit the most complex

---

[4]Bot complexity categorizations are explained in App. C.

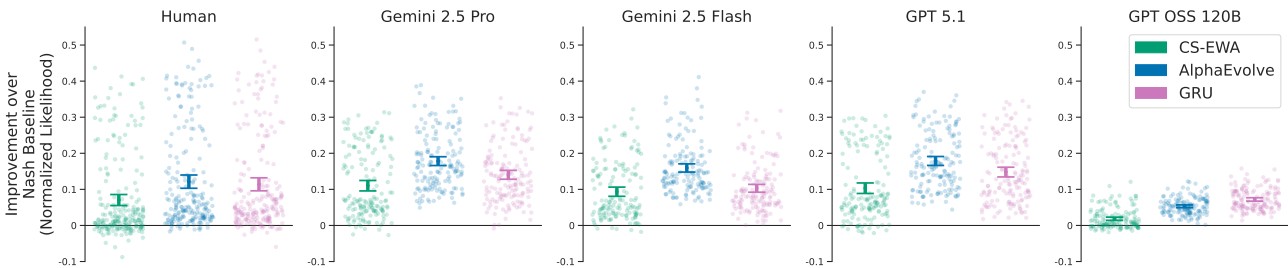

*Figure 3.* **AlphaEvolve improves over baseline behavioral models.** The improvement in *per-game* normalized likelihood over the Nash baseline (which achieves a likelihood of $1/3$ on all datasets) is shown for each dataset and method. Each point corresponds to the normalized, two-fold cross-validated likelihood score of a behavioral model on a single game. The mean and 95% confidence interval are displayed in the error bars.

bot. In a similar spirit, the win rates against nonadaptive bots are largely higher than for the more complex adaptive bots. Thus, **frontier LLMs outperform humans, but overall aggregate win rate trends are similar**.

Differences between humans and LLMs occur in *which* bots they perform especially poorly or well against. While all achieve near-random win rates against the most complex bot, strikingly, humans and GPT OSS 120B *also* perform at near-random levels for the two next most complex bots. On the other hand, Gemini 2.5 Flash/Pro and GPT 5.1 have overall higher win rates than humans, improving in particular against the two aforementioned bots.

Next, we consider the win rates over time (bottom row), which provide two interesting insights. First, Gemini 2.5 Flash/Pro and GPT 5.1 converge to a near-optimal win rate *sooner* than humans do, demonstrating that **advanced frontier models identify strategic patterns in the bot behavior much more rapidly than humans do**, and likely explaining why they have higher aggregate win rates. Yet, the win rate at convergence is similar for humans and the stronger LLMs, suggesting that humans also eventually learn to exploit strategic regularities. Second, GPT OSS 120B's win rates actually *decrease* over time, demonstrating a substantial difference from other agents. We hypothesize that GPT OSS 120B is unable to effectively synthesize the information in longer-context scenarios. Our findings align with prior work, which has also found that weaker LLMs fail to behave in strategically sensible ways (Qian et al., 2025b; Fan et al., 2024).

### 4.2. AlphaEvolve Models Improve Over Baselines

To validate our use of AlphaEvolve as a behavioral modeling method, we compare it to the baselines described in Sec 3.3. All methods were fit to the data three independent times and evaluated on a heldout test set. For the RNN, an extensive hyperparameter search was performed using a split of the training set. Since we are interested in discussing the *best* discovered behavioral program in Section 4.3, the results

reported in this section correspond to the best of three trials for all methods. Details on the training and evaluation procedure are provided in App. E.4.

Fig. 3 displays quality-of-fit for all behavioral modeling methods, in terms of the relative improvement in twofold cross-validated normalized likelihood over the Nash equilibrium baseline. Each individual point represents a single game in the test set. For all datasets, all behavioral models predict behavior substantially better than the fully rational Nash baseline, and AlphaEvolve significantly outperforms the CS-EWA baseline (all $p < 0.001$, Wilcoxon signed rank with Bonferroni correction; $Z \leq -7.148$). AlphaEvolve fits the data similarly well to the RNN for the human and GPT OSS 120B datasets, and improves over the RNN for Gemini 2.5 Flash/Pro and GPT 5.1. We attribute this to the fact that RNNs are more prone to overfitting, whereas AlphaEvolve's cross-validated scoring mechanism and the use of LLMs for code generation implicitly regularize the discovered models toward solutions that generalize. Regardless, this demonstrates that **AlphaEvolve has captured the structure in the dataset at least as well as a highly flexible neural network-based method, and much better than the BGT model designed for human data**.

### 4.3. Analyzing AlphaEvolve Programs to Gain Novel Insights on Human versus LLM Behavior

Unlike RNNs, AlphaEvolve's programmatic models are interpretable. Each program is a human-readable Python function, grounded in scientific concepts drawn from BGT, multi-agent learning, and cognitive psychology. Thus, structural differences between human and LLM behavior may be reflected in the mechanisms embedded within programs that fit the data well. Importantly, we do not claim that the programs provide ground-truth, mechanistic accounts of how agents make strategic decisions in IRPS. Rather, we view the programs as mechanistic hypotheses validated by predictive fit to data, and that provide candidate explanations for observed behavior. Applying Occam's Razor, we study not the best-fit programs discussed in Section 4.2, but

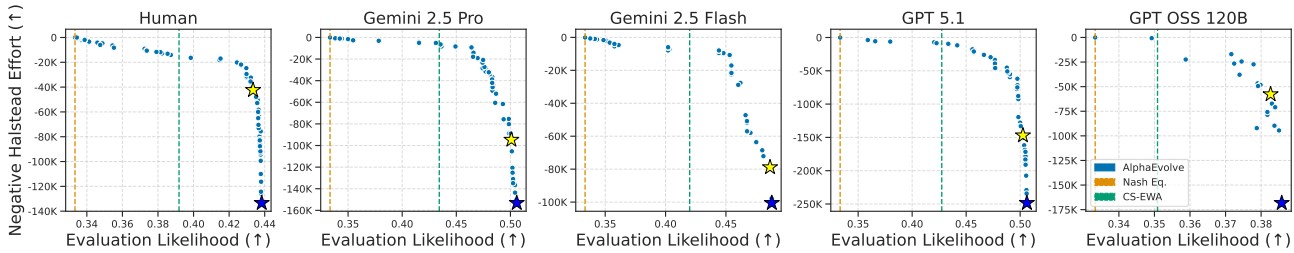

*Figure 4.* **Pareto Frontier of AlphaEvolve Programs.** The AlphaEvolve program evaluated in Fig. 3 is denoted by the blue star, while the simplest-but-best program is indicated by the yellow star. Program simplicity is measured by the negative Halstead effort.

,

the *simplest* programs that fit the data well.

### 4.3.1. SIMPLEST-BUT-BEST DISCOVERED PROGRAMS

Fig. 4 displays the *evaluation score* of all programs located on the Pareto frontier of *training* likelihood and negative Halstead effort. Informally, the Pareto frontier consists of the programs that are not dominated by any other programs; that is, no other program achieves a higher training likelihood while remaining equally or more simple. For each dataset, there are a large number of programs that have similarly high evaluation likelihoods, but vary in complexity. Driven by this observation, we define a straightforward decision rule to identify the *simplest-but-best* (SBB) program from the Pareto frontier.

Let $\ell(\phi)$ denote the evaluation score of program $\phi$, $s(\phi)$ denote its simplicity, $\hat{\Phi}$ denote the set of all programs produced by AlphaEvolve, and $\mathrm{PF}(\hat{\Phi})$ denote the Pareto frontier of simplicity and evaluation likelihood. Then, $\epsilon > 0$, the $\mathrm{SBB}(\epsilon)$ program is defined as:

$$\mathrm{SBB}(\epsilon) \in \arg\max_{\phi \in \mathrm{PF}(\hat{\Phi})} \{s(\phi) \mid \ell(\phi) > \max_{\phi' \in \hat{\Phi}} \ell(\phi') - \epsilon\}. \quad (1)$$

Eq. 1 filters programs on the Pareto frontier for those with evaluation likelihood within $\epsilon$ of the maximum evaluation likelihood, and selects the simplest such program. Our analysis sets $\epsilon$ to $0.005$. $\mathrm{SBB}(0.005)$ programs are displayed as the yellow stars in Fig. 4, while the best-fit programs discussed in Section 4.2 are displayed as blue stars. The code for SBB programs is released in App. F, while Fig. 1 provides schematics of the human and Gemini 2.5 Pro SBB programs. The robustness of SBB programs to variations in the IRPS payoff structure is validated in App. B.2.

### 4.3.2. ABILITY OF SBB PROGRAMS TO PREDICT BEHAVIOR OF OTHER AGENTS

We evaluate each SBB program's ability to predict behavior across agents using the cross-generalization matrix in Fig. 5. Columns represent each SBB program, and rows represent evaluation datasets; note that values are comparable only within rows. Each program's parameters are fit to the evaluation dataset using gradient descent, and cross-validated like-

lihoods are reported. The dominance of diagonal elements confirms that AlphaEvolve identifies programs with high quality-of-fit for each respective agent. High off-diagonal scores, where programs generalize to agents they were not optimized for, reveal behavioral similarities. Symmetric high scores provide stronger evidence of similar strategic behavior between agent pairs.

We find high symmetric predictability between Gemini 2.5 Flash, Pro, and GPT 5.1. Across these models, cross-generalization scores remained consistently high (difference from best-fit model is $0$ for 2.5 Pro dataset, $0.005$ for 2.5 Flash dataset, and $0.017$ for GPT 5.1 dataset), suggesting they behave similarly despite marginal variations in fit. In contrast, GPT OSS 120B's failure to model other agents suggests strategic differences from other agents, and aligns with its declining win rates observed in Section 4.1. Further, a significant performance gap exists between humans and all LLMs: the human program is a poorer predictor of LLM behavior than models optimized for those agents, and vice versa for the human dataset (all $p < 0.001$, Wilcoxon signed-rank with Bonferroni corrections; $Z \leq -5.73$).

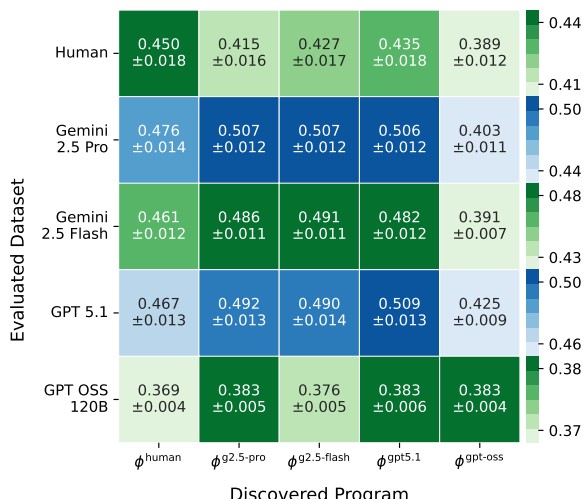

*Figure 5.* **Cross-generalization of SBB Programs to each dataset.** Cross-generalization matrix displaying evaluation likelihoods of each simplest-but-best program on all datasets and 95% CI's.

These results support a core premise of the study: that **LLMs exhibit strategic behaviors that are not fully captured by behavioral models designed for humans**.

### 4.3.3. SEMANTIC CONTENT OF SBB PROGRAMS

We now analyze the semantic content of SBB programs to identify potential mechanistic drivers of the observed behavioral differences between human and LLMs. For brevity, this section focuses on high-level similarities and differences. The properties of each program are summarized in App. Table B.5.

Across all agents, SBB programs consistently exhibit two components, *value-based learning* and *opponent modeling*. Each is described below. Using an LLM judge, we also verify that over the Pareto frontiers of AlphaEvolve programs for humans and Gemini 2.5 Pro, **opponent modeling is a concept included in the majority of programs that fit the data well, while Q-learning is present in nearly all programs.** (App. B.3).

**Value-Based Learning.** All programs but GPT OSS 120B use a 3x3x3 Q-table representing the value of the current action given the prior joint action, $Q(a_t, a_{t-1}^o, a_{t-1})$, while GPT OSS 120B maintains only a single-dimensional Q-vector, representing $Q(a_t)$. These programs update Q-values toward the running average of rewards $r_t$, while applying a decay factor to older rewards.

**Opponent Modeling.** All programs feature an opponent model that tracks dependencies in opponent historical action patterns. Opponent models are used to forecast the opponent's next move, which is used for Q-table based next action selection. Since these models directly model opponent choice frequencies and do not recursively consider the agent itself, all agents are characterized by their respective SBB programs as *level-1* players in cognitive hierarchy theory (Camerer et al., 2004).[5] Since the exploitative adaptive bots are level-1 players, this reflects the limited ability of all agents to counter-exploit adaptive bots. Beyond these core components, choice stickiness (previously observed in human IRPS behavior by Eyler et al. (2009)) emerged as the most frequent cognitive bias, appearing in the human, Gemini 2.5, and GPT OSS 120B programs.

The primary difference between agents lies in the *dimensionality* of the opponent modeling matrix. The human and GPT OSS 120B programs exhibit single-dimensional opponent models that directly track opponent historical move frequency. In contrast, Gemini 2.5 Flash/Pro maintain 3x3 models that track the opponent's move frequencies conditioned on the previous move, and GPT 5.1 maintains a 3x3x3 model. This suggests that **the superior win rates of Gemini 2.5 Flash/Pro and GPT 5.1 stems from main-taining more complex models of opponent behavior than humans**.

---

[5] A level-1 player plays assuming that the opponent does not model them in return (that is, the opponent is a *level-0* player).

## 5. Discussion

As LLM agents are increasingly integrated into society (Tomasev et al., 2025), detecting and understanding strategic divergences from human behavior is critical. This paper explores structural differences between human and LLM behavior in Iterated Rock-Paper-Scissors (IRPS) by comparing interpretable behavioral models discovered directly from data using AlphaEvolve, a state-of-the-art method for program synthesis. Our work exists in the context of a large body of prior work evaluating LLMs as human proxies and strategic players, and developing models to characterize human behavior, which we explore in App. A (Related Work). Nevertheless, to the best of our knowledge, this represents the first application of automated model discovery tools for behavioral game theory, and for characterizing differences between LLM and human strategic behavior.

Our experiments reveal that in IRPS, frontier LLMs exhibit more sophisticated strategic thinking than humans. In particular, frontier models (Gemini 2.5 Pro, Gemini 2.5 Flash, and GPT 5.1) exhibit significantly higher aggregate win rates than humans by identifying and exploiting opponent strategic patterns much earlier in the interaction. The programmatic behavioral models discovered by AlphaEvolve suggest that while humans and frontier LLMs utilize value-based learning and opponent modeling, frontier models employ substantially larger opponent models. In contrast, smaller, open-source models like GPT OSS 120B perform worse than humans, declining in performance as game history grow. This highlights a limitation in long-context reasoning for smaller architectures.

Remarkably, our findings differ substantially from those of Fan et al. (2024), who studied a similar, 10-round IRPS setting where LLMs were pitted against bots. They found that GPT 3 and GPT 3.5 could perform no better than random play because they could not synthesize any insights about opponents from IRPS action history, while GPT 4 was capable of basic opponent modeling, but remained worse than humans. This reflects the significant advance in LLM capabilities between the previous and current generations. Our findings also have implications for advances in the ToM capabilities of LLMs. Strachan et al. (2024) found that the same GPT-4 model studied by Fan et al. (2024) exhibited ToM abilities on par with those of humans in classic test scenarios, in contrast to earlier LLMs who failed such tests (Sap et al., 2022). Although the scope of this study was limited to ToM in IRPS (opponent modeling), the clear advance in opponent modeling capabilities leads us to hypothesize a similar advance in general ToM abilities.

Our findings echo the growing body of evidence that, while aggregate measurements of LLM behavior can display similar trends to those of human behavior (Xie et al., 2024; Horton, 2023), frontier LLMs are not proxies for human behavior, especially in decision-making scenarios (Gao et al., 2024; Fan et al., 2024; Qian et al., 2025b; Jia & Yuan, 2024). As such, the findings of research studies that rely on LLMs as digital twins, or user simulators, must be interpreted with caution.

This study has demonstrated that AlphaEvolve can provide structural insights on the behavioral differences between LLMs and humans. Unlike existing Chain-of-Thought monitoring techniques (Korbak et al., 2025), our modeling method does not require relying on a model's reasoning traces, which do not always accurately reflect behavior (Chen et al., 2025; Kovarik et al., 2025). Thus, the automated behavioral modeling studied here could be a valuable addition to the existing toolbox of monitoring methods.

**Limitations and Future Work**    This study focuses on descriptive models of the intrinsic strategic behavior of LLMs in IRPS, finding several differences from humans. The models describe *average* human behavior on IRPS, which does not account for individual differences in IRPS ability. It is likely that expert humans would display more sophisticated opponent modeling abilities as well, although it may still differ from the particular type of opponent modeling displayed by LLMs.

The methods and findings of this study suggest interesting lines of future work in the areas of alignment, interpretability, and behavioral game theory. For instance, future work might explore techniques to better align LLM strategic behavior with that of humans. Future work might also assess whether the discovered opponent modeling and value-based learning mechanisms correspond to internal computational processes of LLMs, using methods from mechanistic interpretability such as internal state probing or logit analysis. Finally, this study has demonstrated the feasibility of using AlphaEvolve to learn behavioral models on a single game. Future work might consider using AlphaEvolve to learn *general* models of LLM behavior, that describe behavior over a broad set of games or scenarios.

## Impact Statement

This paper demonstrates that automated behavior modeling tools can enable a deeper understanding of LLM capabilities within interactive scenarios, directly benefiting AI interpretability and alignment research. Our research demonstrates that LLMs are not perfect "digital twins" or proxies for human strategic behavior. Researchers must account for these structural divergences to avoid generating data that misrepresents actual human populations. More broadly, our findings demonstrate that frontier LLMs can effectively integrate behavior patterns into insights about teammate or opponent behavior. Thus, it may be possible to develop LLMs that are more effective and intuitive social partners for humans in the future. However, the same capacity for sophisticated opponent modeling suggests a potential risk for LLMs to out-maneuver human counterparts in strategic social interactions, such as negotiations. As LLMs become increasingly integrated into the economy and everyday life, our work demonstrates that AlphaEvolve can serve as a vital tool for monitoring LLM capabilities, ensuring they remain safely aligned with human expectations.

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

# A. Related Work

This section contextualizes our work with respect to existing literature on LLMs as human proxies, LLMs as strategic players, and symbolic models for describing human decision-making. For a discussion of the literature on human behavior in RPS, see Section 2.

**LLMs As Human Proxies**    The social sciences center around understanding human behavior at the individual, group, and societal levels. A core challenge is gathering high-quality human data for the study at hand, which is both time-consuming and expensive. The demonstration that LLMs can generate remarkably human-like text in chat interfaces  (OpenAI, 2022), and that communities of such agents can generate realistic social behavior (Park et al., 2022; 2024), has driven an explosion of interest in deploying LLMs as *social simulacra*, or *digital twins* to study human behavior (Gao et al., 2024). Various methods have been proposed, from prompt engineering (Horton, 2023; Kim et al., 2024), to agent architectures (Park et al., 2024; Li et al., 2024; Cross et al., 2025; Mao et al., 2025), to fine-tuning (Jarrett et al., 2025; Binz et al., 2025). This application of LLMs has the potential to revolutionize research in both academia (Filippas et al., 2024; Epstein, 2023; Ziems et al., 2024; Chan et al., 2025) and industry (Amble & Cohen, 2025; Korst et al., 2025).

As recognition of this potential grows, so too has a body of research evaluating behavioral similarities and differences between LLMs and humans. A prominent evaluation strategy is the *Turing experiment*, in which a classic experiment is replicated with LLMs, and the results compared to humans' (Aher et al., 2023; Xiao & Wang, 2025; Jia & Yuan, 2024; Capraro et al., 2025). Other studies compare LLMs to humans along cognitive measures (Niu et al., 2024; Coda-Forno et al., 2024), strategic traits such as risk preference (Jia et al., 2024; Xiao & Wang, 2025), rationality (Fan et al., 2024), trust (Xie et al., 2024), bias (Qian et al., 2025a), or theory of mind (Strachan et al., 2024). The results from these studies are mixed, with the overall picture being that LLM behavior is often, but not always (e.g. Gao et al. (2025)) similar to humans, where the degree of similarity varies per-scenario and per-LLM. Unlike existing literature, this study leverages a model discovery tool to automatically explore structural differences between humans and LLMs, without relying on pre-existing models and measures of human behavior.

**LLMs As Strategic Players**    There has also been interest in understanding intrinsic strategic capabilities of LLMs (Cipolina-Kun et al., 2025; Sun et al., 2025). Prior studies have examined RPS (Akata et al., 2025; Fan et al., 2024; Lanctot et al., 2023), negotiation games (Qian et al., 2025b; Bianchi et al., 2024), Prisoner's Dilemma (Fontana et al., 2025; Roberts et al., 2025), and market games (Jia & Yuan, 2024). Studies have also examined how LLMs strategically interact with other LLMs in tournament-style evaluations (Payne & Alloui-Cros, 2025; Akata et al., 2025). Most of the aforementioned papers focus solely on the strategic behavior of LLMs, but Qian et al. (2025b) and Roberts et al. (2025) also compare human and LLM strategic behavior in negotiation games and prisoner's dilemma, respectively. In contrast to these works, we compare human and LLM behavior in IRPS. Further, our model-based comparison allows us to compare behavior on a structural level.

**Symbolic Models for Human Decision-Making**    Symbolic models have been used in neuroscience and cognitive science since the mid-1900s to predict how humans or animals make decisions, by describing underlying cognitive mechanisms (Daw, 2011; Disner et al., 2011; Corrado & Doya, 2007; Beck, 1967). Symbolic models have also played a foundational role in behavioral game theory and economics (Camerer, 2003; 2014) (although classically, the focus has been on describing behavior, rather than proposing cognitively plausible theories). For instance, cognitive hierarchies (Camerer et al., 2004), level-$k$ reasoning (Stahl & Wilson, 1995), and quantal response equilibria (McKelvey & Palfrey, 1995) are all prominent models of human strategic decision-making within normal form games. Classically, these models are developed by human researchers, who iteratively refine the models based on scientific theories, principles, and experiments, until they fit the data well. This process is both challenging and time consuming. Moreover, the resultant models often do not maximize predictive fit. As a result, researchers turned to machine learning methods, such as neural networks, to either (1) directly fit behavior (Hartford et al., 2016) or (2) augment existing symbolic models in a hybrid approach (Zhu et al., 2025; Miller et al., 2023). Yet the first approach leads to black-box models that do not offer behavioral insights, while the second remains fundamentally constrained in predictive fit by the structure of the base symbolic model.

To address the above limitations, recent papers have used LLMs to directly discover symbolic models from data (Castro et al., 2025; Rmus et al., 2025; Aygün et al., 2025) by combining an LLM's ability to write code and substantial, embedded domain knowledge with an iterative model-fitting framework. These papers have successfully created programs that fit the data as well as neural network baselines while remaining highly interpretable. Within game theory, a similar approach was used to automatically design interpretable mechanisms (Liu et al., 2025). This paper demonstrates that such an approach

can also be used to discover symbolic *behavioral* models directly from data, and extracts insights about human and LLM behavior by comparing the models.

## B. Supplemental Results

### B.1. Win Rates of Adaptive Bots

Figure 6 presents the win rates of human and LLM agents against each adaptive bot. Mirroring the results reported in the main paper for nonadaptive bots, the overall aggregate trends remain consistent between humans and models. Gemini 2.5 Flash/Pro and GPT 5.1 outperform humans against all adversaries, and demonstrate a more rapid learning trajectory.

The main difference is that the overall win rates are lower than against nonadaptive bots, with the humans' and GPT OSS 120B win

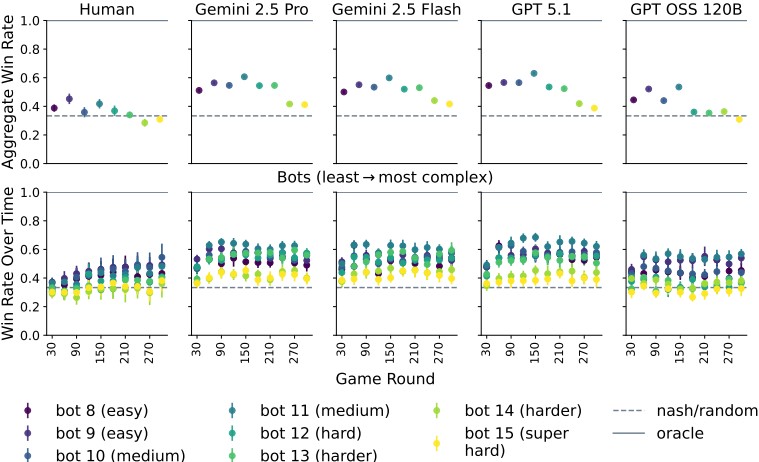

*Figure 6.* **Win rates against adaptive bots.**

rates even dropping below that of the Nash equilibria for the most challenging two bots, demonstrating that the bots are successfully exploiting the aforementioned agents. However, **Gemini 2.5 Flash/Pro and GPT 5.1 are not exploited by the same bots** (although they are performing only slightly better than the Nash equilibria).

### B.2. Robustness of SBB Programs to Variations in Payoff Matrix

We validate the robustness of the SBB programs to minor variations in game payoff structure by assessing the Gemini 2.5 Flash SBB program on gameplay data where IRPS payoffs are multiplied by 10. We find that the SBB program achieves a cross-validated likelihood of 0.462 under this variant, compared to the SBB program likelihood on the original dataset of 0.491, representing a relatively minor decline.

### B.3. Prevalence of Opponent Modeling and Q-Learning Across Pareto Frontier

Opponent modeling and Q-learning are two core components within all SBB programs. A natural question then, is whether these ideas are conceptual outliers, or from another perspective, whether AlphaEvolve might have generated other ideas that fit the data equally as well. We analyze the Pareto frontier of programs for humans and Gemini 2.5 Pro to determine the prevalence of these ideas. For each program, we asked Gemini 2.5 Flash to summarize the core and secondary ideas within each program, providing a short description and justification of each idea. The prompt is shown in Listing 2. Key phrase matching was performed across summaries for opponent modeling and Q-learning.

Fig. 7 shows that for the Pareto frontiers of both datasets, **opponent modeling is a concept included in the majority of programs that fit the data well, while Q-learning is present in nearly all programs.** This consistency suggests that AlphaEvolve commonly identifies these as important primitives for modeling human and LLM behavior, rather than idiosyncratic features of the selected SBB programs.

*Listing 2.* **Conceptual extraction prompt.** The prompt used to summarize concepts contained in Pareto frontier programs. The full prompt begins with the background information displayed in Listing 5 to ground conceptual extraction.

```
... [Background information, beginning "You are a renowned expert in behavioral game
    theory..."] ...

# Current program

Here is a candidate program you have written.

>> BEGIN CANDIDATE PROGRAM
{code}
>> END CANDIDATE PROGRAM

# Task
```

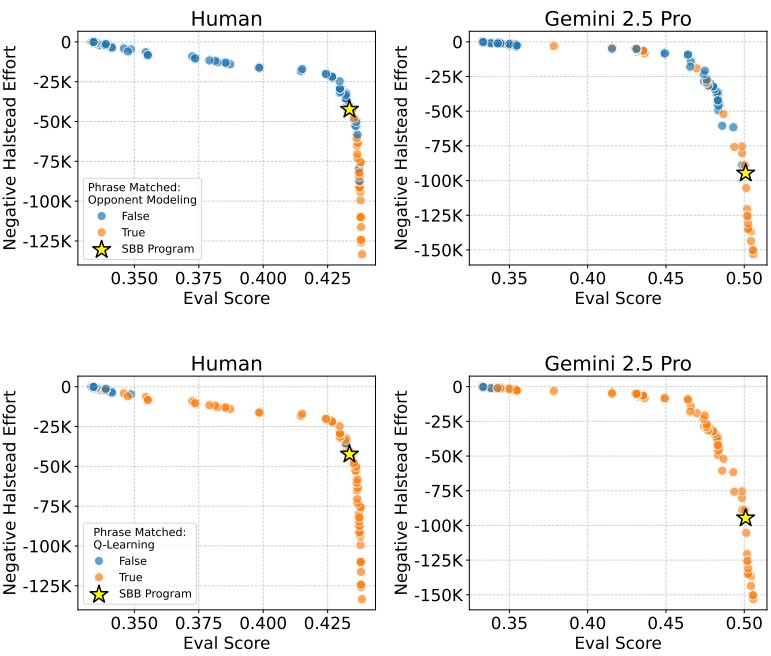

*Figure 7.* **Conceptual classification of AlphaEvolve programs.** Programs on the Pareto frontier containing opponent modeling (*top*) and Q-learning (*bottom*).

```
Given your expertise and familiarity with the literature, please list the distinct ideas
    present in the program, identifying core ideas and secondary ideas.
For each idea, identify whether it may be **ablated** from the program in a scientific
    sense.
This means that the idea may be removed from the program OR replaced by a comparable but
    SIMPLER idea.

## Constraints

- Each idea must represent only a SINGLE concept. If there are two similar ideas (e.g. two
    types of learning rates), list them as separate ideas.

## Answer Format

Explain your decisions but ultimately return your answer with the following format.
You MUST INCLUDE the begin and end tags in your response.

{begin_answer_tag}
  - Core Idea: <name>;
    - Justification: <justification>
    - Ablatable: <true/false>
      - Justification: <justification>
  - Secondary Idea: <name>
    - Justification: <justification>
    - Ablatable: <true/false>
      - Justification: <justification>
{end_answer_tag}
"""
```

## B.4. Offline Evaluation of SBB Program Choice Distributions

We evaluated the extent to which SBB programs capture the statistical properties of the original datasets, specifically focusing on offline win rate replication when conditioned on identical game scenarios. In this evaluation, we replayed

historical games one round at a time, feeding the game state into the SBB programs and sampling an action from the resulting choice logits. These synthetic decisions were then assessed against the ground-truth actions to derive performance metrics. Figure 8 displays the *aggregate* synthetic win rates generated by AlphaEvolve compared to the ground truth, while Figures 9 and 10 provide the breakdown by bot and over time.

We observe that the synthetic win rates align closely with the ground truth, exhibiting particularly high fidelity for nonadaptive bots. While prediction variance naturally increases for adaptive bots and GPT OSS 120B—reflecting the higher entropy inherent in their win rates—the results collectively demonstrate that the choice distributions learned by the SBB programs yield performance profiles comparable to those of the original agents.

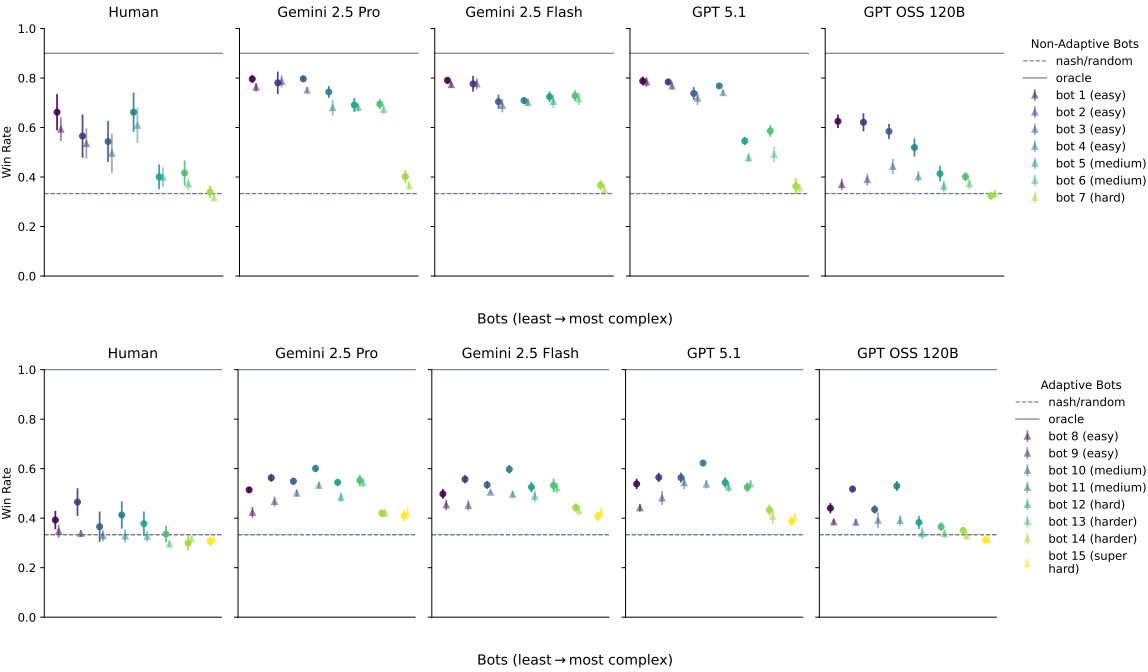

*Figure 8.* **Fidelity of synthetic win rate statistics.** Comparison of ground-truth win rates versus synthetic win rates derived from offline SBB choice distributions. Results are shown for nonadaptive bots (*top*) and adaptive bots (*bottom*).

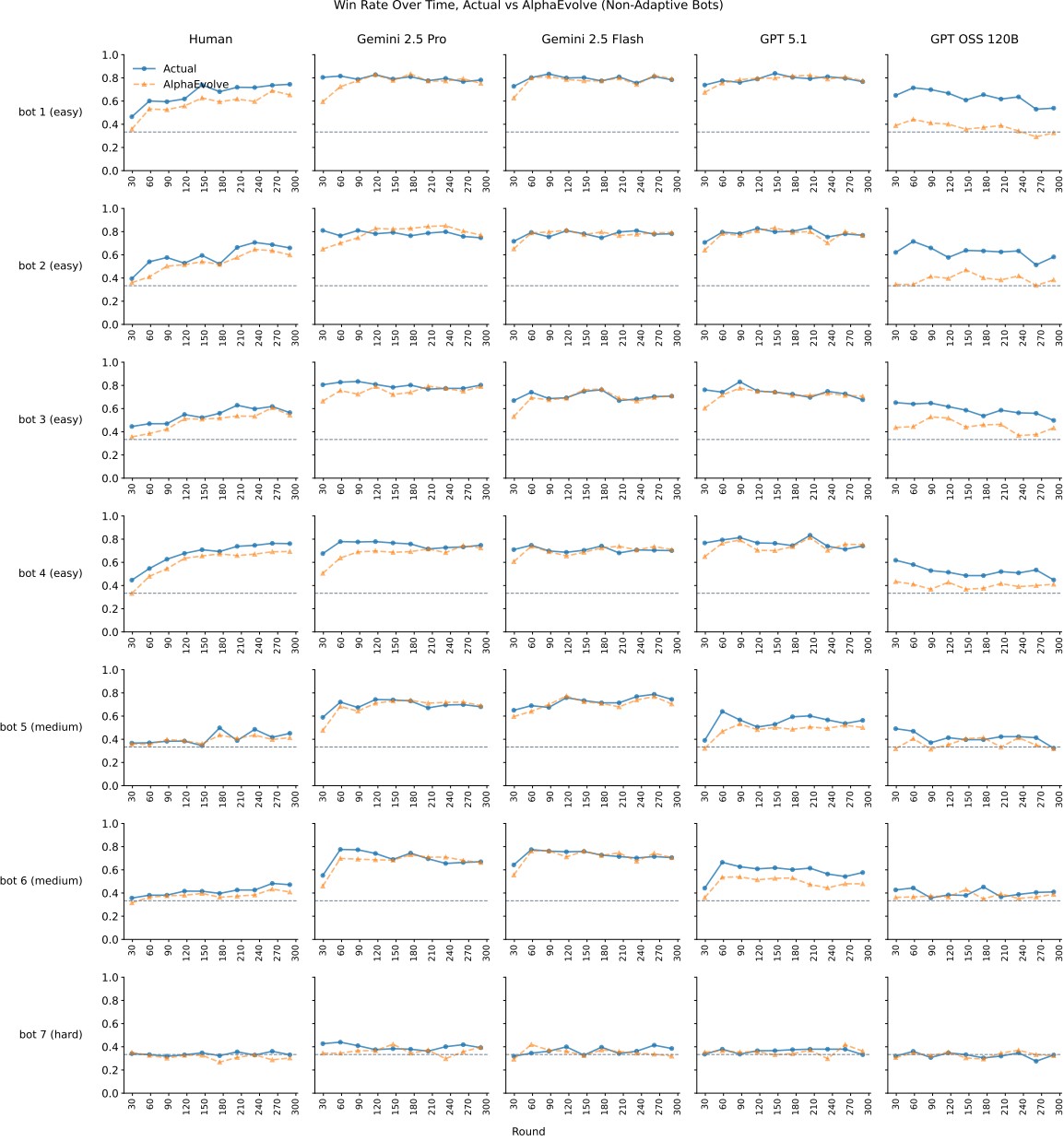

*Figure 9.* **Fidelity of synthetic win rate over time statistics–nonadaptive bots.** Comparison of ground-truth win rates versus synthetic win rates derived from offline SBB choice distributions, broken down per bot.

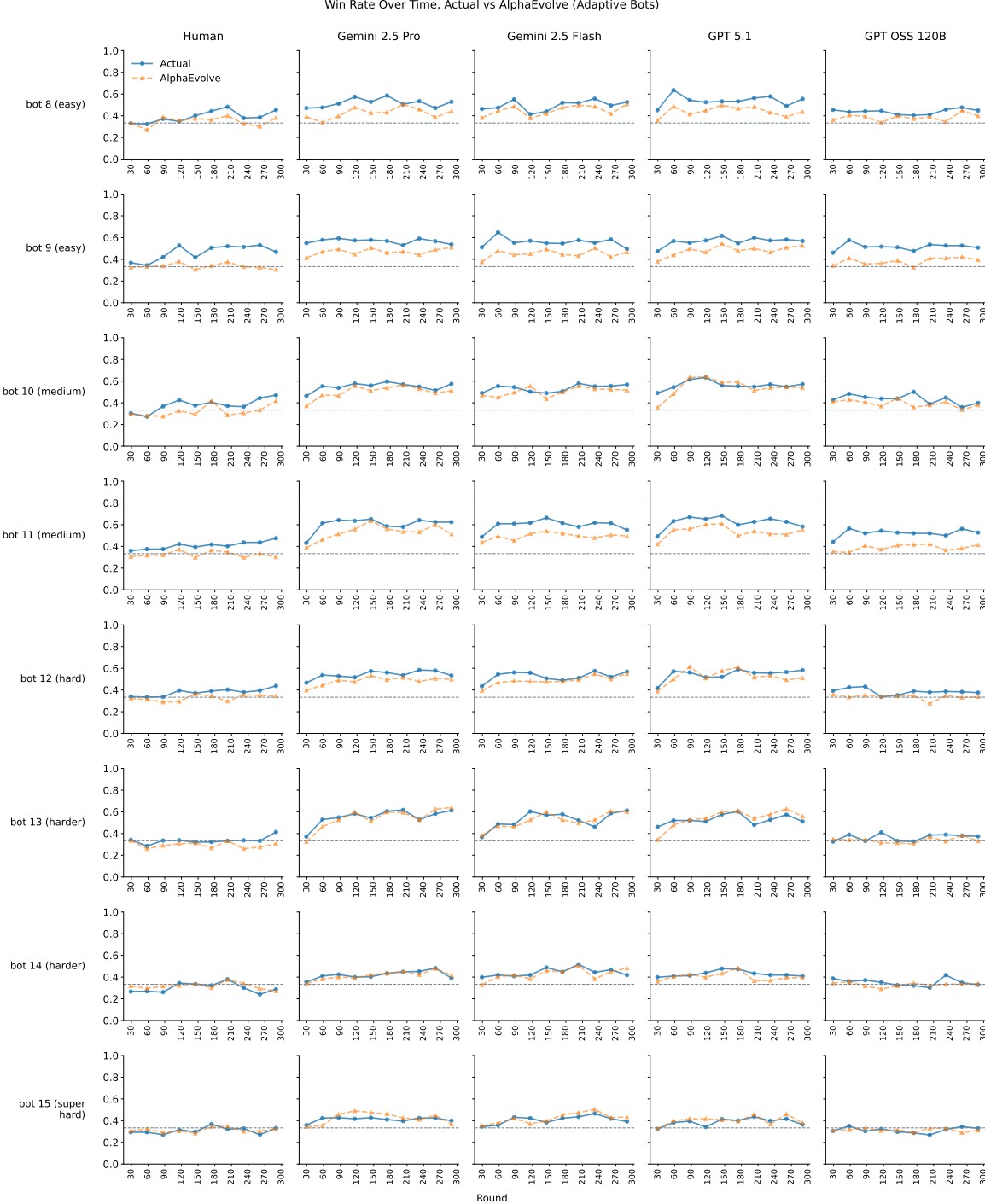

*Figure 10.* **Fidelity of synthetic win rate over time statistics–adaptive bots.** Comparison of ground-truth win rates versus synthetic win rates derived from offline SBB choice distributions, broken down per bot.

### B.5. Simplest-But-Best Programs: Extended Discussion

The SBB programs are included in App. F. Comments were manually pruned to improve code readability. Table 2 summarizes the characteristics of the SBB programs.

## C. IRPS Bots

We implement the bots originally described by Brockbank & Vul (2024). Bots are summarized in Tables 3 and 4. For clarity, we index these bots from 1 to 15, assigning indices 1–7 to nonadaptive agents and indices 8–15 to adaptive agents. The agents are ordered roughly by complexity, which we define based on the quantity of historical information required to update the agent's internal state and compute a choice at timestep $t$. It is important to note that the nonadaptive and adaptive classes differ fundamentally, and our complexity categorization is only meaningful *within* each bot class. Adaptive bots explicitly maintain an opponent model to exploit predictable patterns, whereas nonadaptive bots follow fixed rules mapping history to next action choices. One cannot infer that a low-complexity adaptive bot is strictly more capable than a high-complexity nonadaptive bot, nor that they present an equivalent challenge level, only that adaptive bots are in general more complex than nonadaptive bots.

To formalize the bot policies, we employ the terminology of *positive*, *negative*, and *nil* transitions. We encode the actions Rock, Paper, and Scissors as integers $0, 1$, and $2$ respectively. A transition is defined as *positive* if the agent moves from action $i$ to $(i + 1) \pmod 3$, *negative* if it moves to $(i - 1) \pmod 3$, and *nil* if it repeats action $i$. These transitions can be calculated relative to the bot's own previous action (a self-transition) or relative to the opponent's previous action (an opponent-transition).

Regarding initialization and edge cases, bots act uniformly at random until sufficient history is available to satisfy their specific strategy requirements (e.g., a bot relying on actions from two timesteps ago will play randomly for the first two rounds). Furthermore, in scenarios where an adaptive bot's internal model deems multiple opponent moves equally probable, the bot resolves the ambiguity by sampling one of the predicted moves at random and selecting the corresponding counter-strategy.

## D. Dataset Details

### D.1. Human Dataset

We utilize the human-bot interaction dataset collected by Brockbank & Vul (2024). The dataset consists of human participants playing against either nonadaptive or adaptive bots. The distribution of games across different bot strategies is summarized in Table 11. For full details regarding the data collection procedure, we refer the reader to the original manuscript. The complete instructions provided to the human participants, dataset, and associated code are publicly available at `https://github.com/erik-brockbank/rps-bot-manuscript-public`.

To ensure reproducibility, we applied the following preprocessing and cleaning steps to the raw data files (`rps_v2_data.csv` for nonadaptive and `rps_v3_data.csv` for adaptive bots):

**Exclusion Criteria**  Following Brockbank & Vul (2024), we filtered the dataset based on the following quality controls. First, one specific game ID was removed, consistent with the original authors' data processing, as the participant was identified as not trying. Next, games with fewer than 50 rounds were excluded. Finally, games containing more than 10 missing values in player moves were excluded.

**Imputation**  Incomplete games were padded to a fixed length. Missing player moves (`NaN`) were handled in two steps. Where possible, missing moves were deduced logically based on the opponent's recorded move and the round's payoff. Any remaining missing values were imputed as a "Rock vs. Rock" scenario (resulting in 0 points for both players).

| Bot Strategy | Games | Percent |
|---|---|---|
| *Non-adaptive Bots* | | |
| Bot 1 (easy) | 45 | 16.7% |
| Bot 2 (easy) | 40 | 14.9% |
| Bot 3 (easy) | 35 | 13.0% |
| Bot 4 (easy) | 39 | 14.5% |
| Bot 5 (medium) | 37 | 13.8% |
| Bot 6 (medium) | 36 | 13.4% |
| Bot 7 (hard) | 37 | 13.8% |
| *Adaptive Bots* | | |
| Bot 8 (easy) | 25 | 11.0% |
| Bot 9 (easy) | 29 | 12.8% |
| Bot 10 (medium) | 30 | 13.2% |
| Bot 11 (medium) | 33 | 14.5% |
| Bot 12 (harder) | 23 | 10.1% |
| Bot 13 (harder) | 30 | 13.2% |
| Bot 14 (harder) | 27 | 11.9% |
| Bot 15 (super hard) | 30 | 13.2% |

*Figure 11.* Human dataset composition.

*Table 2.* Simplest-but-best program comparison across datasets.

| Category | Name | Description | Human | Gemini 2.5 Pro | Gemini 2.5 Flash | GPT 5.1 | GPT OSS 120B |
|---|---|---|---|---|---|---|---|
| **Value-based learning** | Q-value matrix size | | $3\times3\times3$ | $3\times3\times3$ | $3\times3\times3$ | $3\times3\times3$ | 3 |
| | Target: temporal difference | Target used in the prediction error is the temporal difference target, i.e. $r_t + \gamma \arg\max_{a'} Q(s_{t+1}, a')$. Compared to the next row, includes a max over actions. | - | - | - | - | - |
| | Target: reward $r_t$ | Target used in the prediction error is the reward $r_t$. | ✓ | ✓ | ✓ | ✓ | ✓ |
| | Counterfactual update | Updates are made on the unchosen actions, based on the opponent's actual action. | - | ✓ | ✓ | ✓ | ✓ |
| | Q-value decay | Q-values are decayed with each timestep. | ✓ | ✓ | ✓ | ✓ | ✓ |
| | Asymmetric learning rate based on sign of pred. error | The learning rate in the Q-value update depends on the sign of the prediction error. | ✓ | - | ✓ | - | - |
| | Asymmetric learning rate based on sign($r_t$) | The learning rate in the Q-value update depends on the sign of the reward. | - | ✓ | - | ✓ *Learning rate based on $|r_t|$. | - |
| **Opponent Modeling (OM)** | OM matrix size | All OM's track opponent action frequencies. The size 3 vector always tracks the opponent's historical action frequency. The 3x3 matrix tracks the opponent transition, $T(a^o_t\|a^o_{t-1})$, while the 3x3x3 matrix tracks $T(a^o_t\|a^o_{t-1}, a^o_{t-2})$. | 3 | $3\times3$ | $3\times3$ | $3\times3\times3$ | 3 |
| | Opponent action frequency decay | Opponent action frequencies are decayed with each timestep. | - | - | ✓ | ✓ | - |
| **Opponent Models' Influence on Decision-Making** | Forecast opponent next choice and play best response. | OM is used to forecast the opponent's next action distribution. The best response is accordingly computed and added to the retuned choice_logits. | ✓ | ✓ | ✓ | ✓ | ✓ |
| **Cognitive Biases** | Alternation bias | Bias to switch to an action other than the current action. | - | - | ✓ | - | ✓ |
| | Stickiness (perseverance) bias | Bias to play own last action. | ✓ | ✓ | ✓ | - | ✓ |
| | Switch-to-beat bias | Bias to play the action that beats own last action. | - | ✓ | - | - | - |

*Table 3.* **Non-adaptive bot strategies.** Bot strategies are described from their perspective. The column 'Dependencies' summarizes the information needed for the bot to choose their action and update internal state variables at time $t$.

| Bot ID | Strategy Name | Dependencies | Description |
|---|---|---|---|
| Bot 1 (Easy) | Self transition (+) | $a_t$ | Plays a positive self-transition. |
| Bot 2 (Easy) | Self transition (-) | $a_t$ | Plays a negative self-transition. |
| Bot 3 (Easy) | Opponent transition (+) | $a_t^o$ | Plays a positive opponent-transition. |
| Bot 4 (Easy) | Opponent transition (0) | $a_t^o$ | Plays a nil opponent-transition. |
| Bot 5 (Medium) | Prev. outcome (W0L+T-) | $a_t, r_t$ | Plays a positive, negative, or nil self-transition following a win, loss, or tie, respectively. |
| Bot 6 (Medium) | Prev. outcome (W+L-T0) | $a_t, r_t$ | Plays a nil, positive, or negative self-transition following a win, loss, or tie, respectively. |
| Bot 7 (Hard) | Prev. outcome, prev. transition | $a_{t-1}, a_t, r_t$ | Plays a self-transition conditioned on the previous outcome and previous self-transition. |

*Table 4.* **Adaptive bot strategies.** Bots are described from their perspective. These bots maintain a count-based model of the opponent to predict future actions. Dependencies list the historical information used to update the bot's internal opponent model at time $t$.

| Bot ID | Strategy Name | Dependencies | Description |
|---|---|---|---|
| Bot 8 (Easy) | Self-transition | $a_{t-1}^o, a_t^o$ | Tracks the history of counts of the opponent's positive, negative, and nil self-transitions. |
| Bot 9 (Easy) | Opponent transition | $a_{t-1}, a_t^o$ | Tracks the history of counts for the opponent's positive, negative, and nil opponent-transitions. |
| Bot 10 (Medium) | Previous move | $a_{t-1}^o, a_t^o$ | Tracks the co-occurrence of every pair of opponent moves from one round to the next. |
| Bot 11 (Medium) | Opp. previous move | $a_{t-1}, a_t^o$ | Exploits patterns in opponent moves based on the bot's own previous move rather than the opponent's. |
| Bot 12 (Harder) | Previous outcome | $r_{t-1}, a_{t-1}^o, a_t^o$ | Tracks the opponent's most likely self-transition conditioned on the previous outcome. |
| Bot 13 (Harder) | Prev. move, Opp. prev. move | $a_{t-1}, a_{t-1}^o, a_t^o$ | Tracks opponent moves conditioned on the joint history of the opponent's and the bot's previous moves. |
| Bot 14 (Harder) | Previous two moves | $a_{t-2}^o, a_{t-1}^o, a_t^o$ | Tracks the opponent's move choices conditioned on the sequence of moves in the previous two rounds. |
| Bot 15 (Super Hard) | Prev. outcome, prev. transition | $a_{t-2}^o, a_{t-1}^o, a_t^o, r_{t-1}$ | Tracks opponent self-transitions conditioned on the previous outcome and the opponent's previous self-transition. |

## D.2. LLM Dataset

As discussed in the main paper, our LLM data gathering procedure was designed to mirror the Brockbank human data collection as closely as possible. Each experimental session consisted of 300 rounds. We collected data for 20 independent sessions for each model against each opponent bot.

Each game was implemented as a series of independent queries to the respective model APIs. For each round $t$ of the game, the agent is provided with a prompt containing the game rules and the complete history of the previous $t - 1$ rounds. The agent is then asked to generate a move for the current round. The exact prompt template is provided in Fig. 3.

All models were accessed using their respective APIs, with their default levels of thinking, from November 2025 through December 2025. For Gemini, this corresponds to the *dynamic thinking* mode; for GPT models, this corresponds to the *medium* level of effort. For all models, we used a temperature to $0.5$. '

*Listing 3.* Prompt used for the LLM Agent in IRPS data gathering.

```
You are playing a game of Rock, Paper, Scissors against an opponent.
In each round, you will select one of the rock, paper, or scissors cards to play against your opponent.
Your opponent is going to choose a card to play as well, but neither of you can see what the other has selected
    until after you have both chosen.

The rules for winning are:
```

```
- Rock beats Scissors
- Scissors beats Paper
- Paper beats Rock

If you and your opponent choose the same card, it's a tie.

You and your opponent are going to play 300 rounds.

Now, it's your turn. The game history is displayed below.

Round 1
Your choice: rock
Opponent choice: scissors
Game outcome: win

Round 2
Your choice: paper
Opponent choice: paper
Game outcome: tie

... [History continues for t-1 rounds] ...

Round t
Based on the history, consider the opponent's strategy and decide on the best move for the current round.
When asked to select your choice for the current round, only output your choice as 'Your choice: rock', 'Your choice
    : paper', or 'Your choice: scissors'.
Do not output any other symbols, thoughts, or text.
```

# E. Behavioral Modeling Procedure

This section describes the details of our behavior modeling procedure. We first describe the general training and evaluation procedure, including the data partitioning procedure, training objective, and evaluation process. Next, we provide details on each baseline and AlphaEvolve. The overall framework for discovering and evaluating behavioral models is shown in Fig. 12.

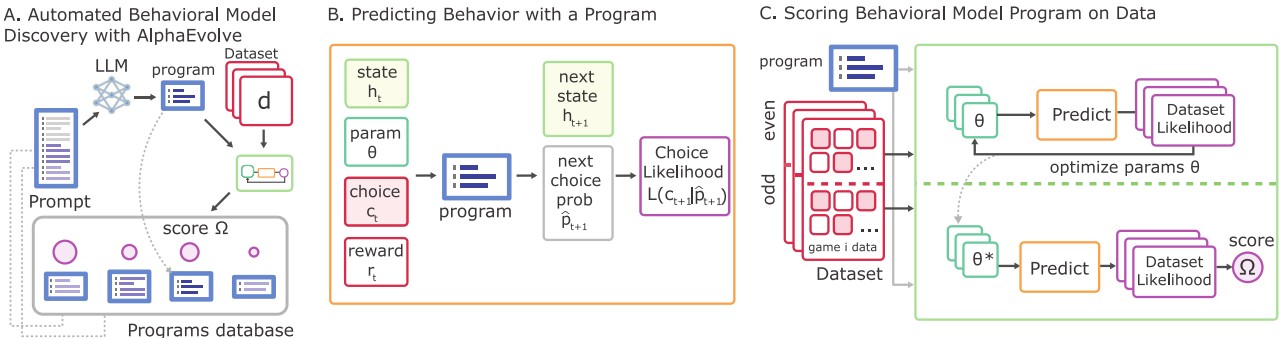

*Figure 12.* **Framework for discovery and evaluation of behavioral models.**

## E.1. Training and Evaluation Procedure

This section describes the general procedure for partitioning the data for training and evaluation, followed by the specific details of the training and evaluation procedure.

**Data Partitioning**    To facilitate rigorous evaluation of model discovery, we partition each dataset into two disjoint sets: a training set $\mathcal{D}_{\text{train}}$ and an evaluation set $\mathcal{D}_{\text{eval}}$. These sets are formed by splitting the data by game index, assigning even-indexed games to the evaluation set and odd-indexed games to the training set. We perform the program search (AlphaEvolve) exclusively on $\mathcal{D}_{\text{train}}$. The output of this training procedure is a set of discovered programs, $\hat{\Phi}$, each endowed with fittable parameters $\theta$. The held-out evaluation set $\mathcal{D}_{\text{eval}}$ is reserved strictly for the final assessment of these discovered programs (which itself involves fitting the behavioral model to data, as described below).

**Behavioral Model Training Procedure**    A behavioral model $\phi(\theta)$ is *fit* to a dataset $\mathcal{D}$ by solving the following *maximum likelihood estimation* objective, where recall that $N$ is the total number of games and $T$ is the number of rounds within each

game:

$$\theta^* = \arg \min_{\theta} \mathcal{L}(\theta, \mathcal{D}; \phi) \tag{2}$$

$$\text{s.t. } \mathcal{L}(\theta, \mathcal{D}; \phi) := \sum_{i=1}^{N} \sum_{t=1}^{T-1} - \log \Pr(a_{i,t+1} | a_{i,t}, a_{i,t}^o, r_{i,t}, h_{i,t}, \theta) \tag{3}$$

$\mathcal{L}(\theta, \mathcal{D}; \phi)$ represents the *negative* log likelihood of the program $\phi(\theta)$ on $\mathcal{D}$, so we *minimize* it over the dataset.

Note that we fit parameters to the *aggregate* population of subjects rather than per-subject. This is due to the structure of the human dataset, where each subject typically completes only a single game. Fitting parameters per-subject would lead to overfitting. However, note that even if we had access to a human dataset where each subject completes multiple games, we should still fit only a single set of parameters for each LLM dataset, since each LLM dataset represents the behavior of a single LLM without persona-based scaffolding.

To solve the objective in Eq. 2, we utilize stochastic gradient descent. Following Castro et al. (2025), we use the AdaBelief optimizer with a learning rate of $5 \times 10^{-2}$. Optimization proceeds until convergence, defined as a relative change in score of less than $10^{-2}$ over a 100-step window, or until a maximum of 10,000 steps is reached. To mitigate sensitivity to initialization and avoid local minima, we repeat this parameter fitting process from distinct random initializations up to 10 times and select the parameters yielding the lowest negative log-likelihood.

### E.2. Behavioral Modeling Evaluation Procedure

All results reported in the main paper are derived from the held-out evaluation set $\mathcal{D}_{\text{eval}}$. To evaluate a program $\phi$, we perform two-fold cross-validation on $\mathcal{D}_{\text{eval}}$. The games in $\mathcal{D}_{\text{eval}}$ are randomly partitioned into two folds, $A$ and $B$. We fit a single set of aggregate parameters $\theta_A$ to the games in fold $A$ and evaluate the negative log-likelihood on fold $B$. Conversely, we fit $\theta_B$ to fold $B$ and evaluate on fold $A$. The final reported metric is the average normalized likelihood across these folds.

### E.3. Baselines

**Contextual Sophisticated Experience Weighted Attraction (CS-EWA)**   We introduce Contextual Sophisticated Experience Weighted Attraction (CS-EWA), a baseline that extends the classic EWA model (Camerer & Hua Ho, 1999) to account for temporal dependencies and strategic sophistication in repeated interactions. While standard EWA and its sophisticated variant (Camerer et al., 2002) assume random rematching (mean-field interaction), our CS-EWA agent explicitly models the history of the two-player interaction.

The model conditions its learning on a joint history state $S_t$. Let $a_t$ and $a_t^o$ denote the agent's and opponent's actions at time $t$, respectively. For a history length $L$, the state at time $t$ is defined as the sequence of the previous $L$ joint actions: $S_t = (a_{t-L}, a_{t-L}^o, \ldots, a_{t-1}, a_{t-1}^o)$. The agent maintains two independent sets of EWA learning tables, $\mathcal{T}_{\text{self}}$ and $\mathcal{T}_{\text{shadow}}$, each containing $K^{2L}$ entries (where $K$ is the number of actions). $\mathcal{T}_{\text{self}}$ tracks the agent's own history of payoffs to form an *adaptive* policy, while $\mathcal{T}_{\text{shadow}}$ tracks the opponent's history to form a predictive model of the opponent's behavior.

The decision process at step $t$ proceeds in three phases:

1. **Retrospective Update:** Given the most recent observation $(a_{t-1}, a_{t-1}^o)$ and reward $r_{t-1}$, the agent retrieves the state index $S_{t-1}$ (the context used for the previous decision). It updates the experience weights $N$ and attractions $A$ in $\mathcal{T}_{\text{self}}$ using the standard EWA learning rules. Simultaneously, it updates $\mathcal{T}_{\text{shadow}}$ from the opponent's perspective, treating $a_{t-1}^o$ as the choice and $a_{t-1}$ as the environment.

2. **Strategic Forecasting:** The agent identifies the current state $S_t$. It computes a probabilistic forecast of the opponent's move, $\pi_{\text{opp}}$, as a mixture of two hypothesized behaviors: an *adaptive* opponent (derived from $\mathcal{T}_{\text{shadow}}$) and a *sophisticated* opponent (who best-responds to the agent's own adaptive tendencies). This belief mixture is controlled by a parameter $\alpha' \in [0, 1]$.

3. **Action Selection:** The agent calculates a *sophisticated* strategy (best-responding to $\pi_{\text{opp}}$) and mixes it with its own *adaptive* strategy (derived from $\mathcal{T}_{\text{self}}$), controlled by a population parameter $\alpha \in [0, 1]$.

Algorithm 1 details the complete procedure, while Algorithm 2 defines the standard EWA update algorithm. Following the original presentation of EWA (Camerer & Hua Ho, 1999), $N$ is overloaded to refer to the number of times the context has been seen, while $\alpha, \alpha', \phi, \delta, \rho, \beta$ refer to EWA-specific parameters.

---

**Algorithm 1** Contextual Sophisticated EWA (CS-EWA)

---

1: **Hyperparameters:** $\theta_{\text{self}}, \theta_{\text{shadow}}$ (EWA params: $\phi, \delta, \rho, \beta$), $\alpha$ (sophistication), $\alpha'$ (belief).
2: **Initialize:** State tables $\mathcal{T}_{\text{self}}, \mathcal{T}_{\text{shadow}}$ for all $K^{2L}$ contexts; History buffer $H$.
3: **while** interaction continues **do**
4:     *// 1. Update Phase (Post-Trial $t-1$)*
5:     Observe previous actions $a_{t-1}, a_{t-1}^o$
6:     $S_{prev} \leftarrow \text{Encode}(H)$
7:     *// Update Self Table (Agent perspective)*
8:     $\mathcal{T}_{\text{self}}[S_{prev}] \leftarrow \text{EWA\_Update}(\mathcal{T}_{\text{self}}[S_{prev}], \theta_{\text{self}}, a_{t-1}, a_{t-1}^o, \text{Payoff}_{\text{self}})$
9:     *// Update Shadow Table (Opponent perspective)*
10:     $\mathcal{T}_{\text{shadow}}[S_{prev}] \leftarrow \text{EWA\_Update}(\mathcal{T}_{\text{shadow}}[S_{prev}], \theta_{\text{shadow}}, a_{t-1}^o, a_{t-1}, \text{Payoff}_{\text{opp}})$
11:     Update $H \leftarrow \text{Shift}(H, a_{t-1}, a_{t-1}^o)$
12:     *// 2. Prediction Phase (Trial $t$)*
13:     $S_{curr} \leftarrow \text{Encode}(H)$
14:     Retrieve attractions $A_{\text{self}}$ from $\mathcal{T}_{\text{self}}[S_{curr}]$ and $A_{\text{shadow}}$ from $\mathcal{T}_{\text{shadow}}[S_{curr}]$
15:     *// Component A: Agent's Adaptive Strategy*
16:     $\pi_{\text{self}}^{\text{adapt}} \leftarrow \text{Softmax}(\beta_{\text{self}} \cdot A_{\text{self}})$
17:     *// Component B: Forecast Opponent (Level-k Closure)*
18:     $\pi_{\text{opp}}^{\text{adapt}} \leftarrow \text{Softmax}(\beta_{\text{shadow}} \cdot A_{\text{shadow}})$
19:     $V_{\text{opp}}^{\text{soph}} \leftarrow \mathbb{E}_{a \sim \pi_{\text{self}}^{\text{adapt}}}[\text{Payoff}_{\text{opp}}(\cdot, a)]$
20:     $\pi_{\text{opp}}^{\text{soph}} \leftarrow \text{Softmax}(\beta_{\text{shadow}} \cdot V_{\text{opp}}^{\text{soph}})$
21:     $\pi_{\text{opp}}^{\text{forecast}} \leftarrow \alpha' \pi_{\text{opp}}^{\text{soph}} + (1 - \alpha') \pi_{\text{opp}}^{\text{adapt}}$
22:     *// Component C: Agent's Sophisticated Strategy*
23:     $V_{\text{self}}^{\text{soph}} \leftarrow \mathbb{E}_{a^o \sim \pi_{\text{opp}}^{\text{forecast}}}[\text{Payoff}_{\text{self}}(\cdot, a^o)]$
24:     $\pi_{\text{self}}^{\text{soph}} \leftarrow \text{Softmax}(\beta_{\text{self}} \cdot V_{\text{self}}^{\text{soph}})$
25:     *// 3. Final Mixture*
26:     $\pi_{\text{final}} \leftarrow \alpha \pi_{\text{self}}^{\text{soph}} + (1 - \alpha) \pi_{\text{self}}^{\text{adapt}}$
27:     Sample action $a_t \sim \pi_{\text{final}}$
28: **end while**

---

---

**Algorithm 2** Standard EWA Update

---

1: **Input:** Current State $(N, A)$, Params $(\rho, \phi, \delta)$, Choice $a_{\text{chosen}}$, Opponent Choice $a_{\text{chosen}}^o$, Payoff Function $U(\cdot, \cdot)$
2: **Output:** Updated State $(N', A')$
3: $N' \leftarrow \rho N + 1$
4: **for** each action $a \in \{1 \dots K\}$ **do**
5:     *// Calculate realized or counterfactual payoff from reward matrix U*
6:     $r_a \leftarrow U(a, a_{\text{chosen}}^o)$
7:     *// Indicator for chosen action*
8:     $I_a \leftarrow \mathbb{I}(a = a_{\text{chosen}})$
9:     *// Update attractions.*
10:     $A'_a \leftarrow \frac{\phi N A_a + [\delta + (1-\delta)I_a]r_a}{N'}$
11: **end for**
    Return $(N', A')$

---

**RNN Implementation** To establish a neural baseline, we trained a recurrent neural network that included several Gated Recurrent Unit (GRU) blocks followed by several fully connected layers, where the number of such blocks and layers

were determined via sweeping hyperparameters. For the GRU blocks, we swept the values $\{1, 2\}$; for the fully connected layers, we swept $\{0, 1, 2, 4\}$. We also swept the number of hidden units, considering sizes of $\{8, 16\}$. We additionally swept over learning rates of $\{10^{-3}, 10^{-4}\}$, optimizing all model variants using the Adam optimizer. To ensure robustness, each hyperparameter configuration was trained using 3 distinct random seeds. We utilized early stopping to select the best parameters and terminate training.

Consistent with our overall modeling approach, we fit a single across-game model. All training and hyperparameter selection was conducted using the training split. The best set of hyperparameters were evaluated with three distinct seeds, via two-fold cross-validation. Consistent with how we select the final AlphaEvolve seed, we select the *best* performing RNN seed on the heldout evaluation set.

### E.4. AlphaEvolve Details

**Program Template**  All programs discovered by AlphaEvolve conform to the functional signature defined in Listing 4. The template is implemented in JAX (Bradbury et al., 2018), enabling efficient automatic differentiation. This allows for rapid parameter fitting via gradient descent for every candidate program generated during the evolutionary process. Following Castro et al. (2025), the `params` variable is a vector that is limited to size 10. Examples of programs produced by AlphaEvolve can be found in App. 4.3.3.

```
def agent(
    params: jnp.array,
    choice: int,
    opponent_choice: int,
    reward: float,
    agent_state: Optional[jnp.array],
) -> tuple[jnp.array, jnp.array]:
    """Behavioral model describing agent behavior on iterated rock-paper-scissors."""

    agent_state = None
    choice_logits = jnp.array([1.0, 1.0, 1.0])

    return choice_logits, agent_state
```

*Listing 4.* **Initial template program for AlphaEvolve.** All discovered programs follow this functional signature. The initial program is equivalent to the Nash equilibrium model.

**LLM Prompting**  AlphaEvolve uses a structured prompt to guide the LLM in generating valid program candidates. The full prompt utilized for the two-player scenarios is displayed below.

*Listing 5.* The prompt provided to AlphaEvolve for discovering two-player game strategies

```
You are a renowned expert in behavioral game theory, with deep expertise in psychology, neuroscience, machine
    learning, multi-agent reinforcement learning, and many other related fields. You are also a highly skilled
    software engineer. Leveraging your deep knowledge of scientific literature and your innovative spirit, you
    excel at implementing new ideas for computational models of human behavior in Python and skillfully prototyping
    them.

Your job is to develop candidate models of human behavior, implemented as Python programs, that will be evaluated on
    their ability to reproduce the behavior of humans playing two-player games where they iteratively perform
    actions and learn from the outcomes of their behavior. These programs have parameters that will be fit to the
    behavior of an individual subject, and will be scored based on how well the model reproduces the behavior of
    that same subject in a held-out dataset. They will also be scored based on how understandable they are to a
    fellow scientist.

# Context

## Program structure
The program you are writing will have the name 'agent', and will implement an agent that learns and behaves like the
    subjects do. It will have fittable parameters which allow it to match the behavior of different subjects
    performing the same task. Programs will be implemented in jax, and must be fully differentiable, so that
    parameters can be efficiently optimized when computing the score.

The program will describe the computations that happen within a single trial, and will have the following internal
    structure. First, parameters from the input 'params' jax array will be assigned names. These names should be
    descriptive of their role in the code. Next, the 'state' array will be updated to reflect the subject's
    experience. Finally, the probability of each possible choice will be computed, and expressed in the form of
    logits. The program will output both these logits and the updated state. Each computational step should be
    written on its own line so that the code is clear and easy to understand. Any complex or unusual computations
    should be accompanied by an explanatory comment.

## Prior programs
Previously we found that the following programs performed well on the task at hand, though we believe that it is
    still possible to do better:
```

```
{previous_programs}

## Current program
Here is the current program you are trying to improve (you will need to propose
a modification to it below):

{code}

## *SEARCH/REPLACE block* Rules:

Every *SEARCH/REPLACE block* must use this format:
1. The opening fence: ```python
2. The start of search block: <<<<<<< SEARCH
3. A contiguous chunk of up to 4 lines to search for in the existing source code
4. The dividing line: =======
5. The lines to replace into the source code
6. The end of the replace block: >>>>>>> REPLACE
7. The closing fence: ```

Every *SEARCH* section must *EXACTLY MATCH* the existing file content,
character for character, including all comments, docstrings, etc.

*SEARCH/REPLACE* blocks will replace *all* matching occurrences.
Include enough lines to make the SEARCH blocks uniquely match the lines to
change.

Keep *SEARCH/REPLACE* blocks concise.
Break large *SEARCH/REPLACE* blocks into a series of smaller blocks that each
change a small portion of the file.
Include just the changing lines, and a few surrounding lines if needed for
uniqueness.
Do not include long runs of unchanging lines in *SEARCH/REPLACE* blocks.

To move code within a file, use 2 *SEARCH/REPLACE* blocks: 1 to delete it from
its current location, 1 to insert it in the new location.

Example:
```python
<<<<<<< SEARCH
  f = lambda w, z: (w + z, w + z, w + z)
  return f
=======
  f = lambda w, z: (jax.nn.sigmoid(w+z), w + z, w + z)

  return f
{replace}
```

{lazy_prompt}
ONLY EVER RETURN CODE IN A *SEARCH/REPLACE BLOCK*!

## Task
{task_instruction} {focus_sentence}
{trigger_chain_of_thought}Describe each change with a *SEARCH/REPLACE block*.
```

**Multi-Objective Optimization** A key distinction of our implementation compared to Castro et al. (2025) is the optimization criteria. While Castro et al. (2025) optimized primarily for predictive accuracy (likelihood), we employ AlphaEvolve to perform *multi-objective optimization*. We simultaneously optimize two objectives. The first is the maximum likelihood objective defined in Eq. 2, while the second is the Halstead effort, a measure of program complexity. We found that optimizing Halstead effort improved the program interpretability. AlphaEvolve natively supports multi-objective optimization, allowing us to explore the Pareto frontier between model fit and code complexity.

**Selection and Evaluation Procedure** Our evaluation strategy focuses on identifying the single best scientific hypothesis (program) discovered, rather than evaluating the variance of the search method itself. To do so, we performed three independent AlphaEvolve runs. From each run, we extracted the program with the highest training score on $\mathcal{D}_{\text{train}}$. These top candidates were then evaluated on the held-out evaluation set $\mathcal{D}_{\text{eval}}$. As detailed in Appendix E.1, we applied two-fold cross-validation on the evaluation games to compute a robust likelihood score. The final reported model is the single program that achieved the highest cross-validated score on $\mathcal{D}_{\text{eval}}$ across all three random seeds.

### E.5. Halstead Complexity Metrics

To quantify program complexity, we utilize the Halstead effort measure. The Halstead effort is one of a family of Halstead code complexity measures, all of which measure code complexity based on the total and unique number of operators and operands in code

Let $\eta_1$ be the number of distinct operators, $\eta_2$ be the number of distinct operands, $N_1$ be the total number of operators, and $N_2$ be the total number of operands. The Halstead metrics are defined as follows:

- **Volume** ($V$): $V = (N_1 + N_2) \times \log_2(\eta_1 + \eta_2)$, representing the size of the program.

- **Difficulty** ($D$): $D = \frac{\eta_1}{2} \times \frac{N_2}{\eta_2}$, which measures the difficulty of the program to write or understand.

- **Effort** ($E$): $E = D \times V$. This metric represents the total amount of time required to understand and develop the program.

## F. SBB Programs

```python
def human_vs_bot_irps(
    params: chex.Array,
    choice: int,
    opponent_choice: int,
    reward: float,
    agent_state: Optional[chex.Array],
) -> tuple[chex.Array, chex.Array]:

  positive_pe_learning_rate = jax.nn.sigmoid(params[0])
  negative_pe_learning_rate = jax.nn.sigmoid(params[1])
  inverse_temperature_base = jnp.exp(params[2])
  q_decay_rate = jax.nn.sigmoid(params[3])
  stickiness_bias = params[4]
  opponent_history_learning_rate = jax.nn.sigmoid(params[5])
  win_value = params[6]
  loss_value = params[7]
  tie_value = params[8]
  uncertainty_sensitivity = jax.nn.sigmoid(params[9])

  if agent_state is None:
    agent_state = jnp.zeros(33) # Total size: 27 (Q) + 3 (last choice) + 3 (opponent history)

  q_tensor = agent_state[:27].reshape((3, 3, 3))
  last_agent_choice_one_hot = agent_state[27:30]
  opponent_choice_history = agent_state[30:33]

  current_agent_choice_one_hot = jax.nn.one_hot(choice, num_classes=3)
  current_opponent_choice_one_hot = jax.nn.one_hot(opponent_choice, num_classes=3)

  chosen_q_value = jnp.einsum('i,j,k,ijk->', last_agent_choice_one_hot, opponent_choice_history, current_agent_choice_one_hot, q_tensor
      )

  subjective_reward = jnp.where(reward == 0.3, win_value, jnp.where(reward == -0.1, loss_value, tie_value))
  prediction_error = subjective_reward - chosen_q_value
  learning_rate = jnp.where(prediction_error >= 0, positive_pe_learning_rate, negative_pe_learning_rate)

  updated_chosen_q_value = chosen_q_value + learning_rate * prediction_error

  update_mask = jnp.einsum('i,j,k->ijk', last_agent_choice_one_hot, opponent_choice_history, current_agent_choice_one_hot)
  q_tensor = q_tensor * q_decay_rate + update_mask * (updated_chosen_q_value - q_decay_rate * chosen_q_value)

  opponent_history_modulated_learning_rate = jnp.clip(opponent_history_learning_rate * (1 + jnp.abs(subjective_reward)), 0.0, 1.0)
  opponent_choice_history = (1 - opponent_history_modulated_learning_rate) * opponent_choice_history +
      opponent_history_modulated_learning_rate * current_opponent_choice_one_hot

  q_values_for_next_choice = jnp.einsum('i,j,ijk->k', current_agent_choice_one_hot, opponent_choice_history, q_tensor)
  q_values_for_next_choice = q_values_for_next_choice + last_agent_choice_one_hot * stickiness_bias

  dynamic_inverse_temperature = inverse_temperature_base * jnp.exp(-uncertainty_sensitivity * jnp.abs(prediction_error))
  choice_logits = dynamic_inverse_temperature * q_values_for_next_choice

  agent_state = jnp.concatenate([
      q_tensor.flatten(),
      current_agent_choice_one_hot,
      opponent_choice_history,
  ])

  return choice_logits, agent_state
```

*Listing 6.* Human SBB Program.

```python
def gemini_25_pro_irps(
    params: chex.Array,
    choice: int,
    opponent_choice: int,
    reward: float,
    agent_state: Optional[chex.Array],
) -> tuple[chex.Array, chex.Array]:

    learning_rate_win = jax.nn.sigmoid(params[0])
    learning_rate_loss = jax.nn.sigmoid(params[1])
    inverse_temperature = jax.nn.softplus(params[2])
    decay_rate = jax.nn.sigmoid(params[3])
    counterfactual_learning_rate = jax.nn.sigmoid(params[4])
    perseverance_bias = params[5]
    switch_to_beat_bias = params[6]
    opponent_beater_bias = params[7]
    opponent_learning_rate = jax.nn.sigmoid(params[8])
    opponent_inverse_temperature = jax.nn.softplus(params[9])

    if agent_state is None:
        q_values = jnp.zeros((3, 3, 3))
        opponent_model = jnp.full((3, 3), 1/3.0)
        prev_choices = jnp.array([-1, -1])
        agent_state = jnp.concatenate((q_values.flatten(), opponent_model.flatten(), prev_choices))

    q_values = agent_state[:27].reshape((3, 3, 3))
    opponent_model = agent_state[27:36].reshape((3, 3))
    my_prev_choice = jnp.array(agent_state[36], dtype=jnp.int32)
    opponent_prev_choice = jnp.array(agent_state[37], dtype=jnp.int32)

    q_values_decayed = decay_rate * q_values

    is_first_trial = (opponent_prev_choice == -1)

    current_learning_rate_actual = jnp.where(reward > 0, learning_rate_win, learning_rate_loss)
    prediction_error_actual = reward - q_values_decayed[choice, my_prev_choice, opponent_prev_choice]

    q_update_actual_choice = current_learning_rate_actual * prediction_error_actual
    updated_q_values_actual_choice = q_values_decayed.at[choice, my_prev_choice, opponent_prev_choice].add(
        q_update_actual_choice * (~is_first_trial)
    )

    reward_matrix = jnp.array([
        [0.0, -0.1, 0.3],
        [0.3, 0.0, -0.1],
        [-0.1, 0.3, 0.0],
    ])

    counterfactual_rewards = reward_matrix[:, opponent_choice]
    counterfactual_choices_mask = jnp.arange(3) != choice
    all_choices_current_q_values = updated_q_values_actual_choice[:, my_prev_choice, opponent_prev_choice]
    counterfactual_prediction_errors = counterfactual_rewards - all_choices_current_q_values

    counterfactual_updates = counterfactual_learning_rate * counterfactual_prediction_errors * counterfactual_choices_mask
    updated_q_values = updated_q_values_actual_choice.at[:, my_prev_choice, opponent_prev_choice].add(
        counterfactual_updates * (~is_first_trial)
    )

    choice_logits_q_values = inverse_temperature * updated_q_values[:, choice, opponent_choice]
    choice_logits_q_values = choice_logits_q_values.at[choice].add(perseverance_bias)
    choice_logits_q_values = choice_logits_q_values.at[(choice + 1) % 3].add(switch_to_beat_bias)
    choice_logits_q_values = choice_logits_q_values.at[(opponent_choice + 1) % 3].add(opponent_beater_bias)

    opponent_prediction_error = jnp.eye(3)[opponent_choice] - opponent_model[opponent_prev_choice, :]
    opponent_model_update = opponent_learning_rate * opponent_prediction_error * (~is_first_trial)
    updated_opponent_model = opponent_model.at[opponent_prev_choice, :].add(opponent_model_update)
    predicted_opponent_next_choices = updated_opponent_model[opponent_choice, :]
    expected_rewards_opponent_model = jnp.dot(reward_matrix, predicted_opponent_next_choices)
    choice_logits_opponent_model = opponent_inverse_temperature * expected_rewards_opponent_model

    choice_logits = choice_logits_q_values + choice_logits_opponent_model

    next_agent_state = jnp.concatenate((updated_q_values.flatten(), updated_opponent_model.flatten(), jnp.array([choice, opponent_choice
        ])))

    return choice_logits, next_agent_state
```

*Listing 7.* Gemini 2.5 Pro SBB Program.

```
1  def gemini_25_flash_irps(
2      params: chex.Array,
3      choice: int,
4      opponent_choice: int,
5      reward: float,
6      agent_state: Optional[chex.Array],
7  ) -> tuple[chex.Array, chex.Array]:
8
9      if agent_state is None:
10         q_values_init = jnp.zeros((3, 3, 3))
11         initial_previous_choices = jnp.array([0, 0]) # [prev_opponent_choice, prev_agent_choice]
12         opponent_prediction_init = jnp.zeros((3, 3))
13         agent_state = jnp.concatenate((q_values_init.flatten(), initial_previous_choices, opponent_prediction_init.flatten()))
14
15     q_values = agent_state[:27].reshape((3, 3, 3))
16     prev_opponent_choice = agent_state[27].astype(int)
17     prev_agent_choice = agent_state[28].astype(int)
18     opponent_prediction_state = agent_state[29:].reshape((3, 3))
19
20     learning_rate_positive = jax.nn.sigmoid(params[0])
21     learning_rate_negative = jax.nn.sigmoid(params[1])
22     inverse_temperature = jnp.exp(params[2])
23     decay_rate = jax.nn.sigmoid(params[3])
24     alternation_bias_strength = jax.nn.sigmoid(params[4])
25     counterfactual_learning_rate = jax.nn.sigmoid(params[5])
26     regret_learning_rate = jax.nn.sigmoid(params[6])
27     opponent_learning_rate = jax.nn.sigmoid(params[7])
28     opponent_inverse_temperature = jnp.exp(params[8])
29     opponent_prediction_decay = jax.nn.sigmoid(params[9])
30
31     normalized_reward = (reward + 0.1) / 0.4
32
33     q_values = q_values * decay_rate
34     opponent_prediction_state = opponent_prediction_state * opponent_prediction_decay
35
36     prediction_error = normalized_reward - q_values[choice, prev_opponent_choice, prev_agent_choice]
37     effective_learning_rate = jnp.where(
38         prediction_error >= 0, learning_rate_positive, learning_rate_negative
39     )
40     q_values = q_values.at[choice, prev_opponent_choice, prev_agent_choice].add(effective_learning_rate * prediction_error)
41
42     counterfactual_reward = 1.0
43     counterfactual_prediction_error = counterfactual_reward - q_values[winning_choice_against_opponent, prev_opponent_choice,
            prev_agent_choice]
44     q_values = q_values.at[winning_choice_against_opponent, prev_opponent_choice, prev_agent_choice].add(
45         counterfactual_learning_rate * counterfactual_prediction_error
46     )
47
48     unchosen_actions_mask = 1.0 - jax.nn.one_hot(choice, num_classes=3)
49     neutral_reward = 0.5
50     regret_prediction_error = neutral_reward - q_values[:, prev_opponent_choice, prev_agent_choice]
51     q_values = q_values.at[:, prev_opponent_choice, prev_agent_choice].add(
52         regret_learning_rate * regret_prediction_error * unchosen_actions_mask
53     )
54
55     alternation_bias = (1.0 - jax.nn.one_hot(choice, num_classes=3)) * alternation_bias_strength - \
56                        jax.nn.one_hot(choice, num_classes=3) * alternation_bias_strength
57
58     predicted_opponent_next_choice_probs = jax.nn.softmax(opponent_prediction_state[prev_opponent_choice])
59     actual_opponent_next_choice_one_hot = jax.nn.one_hot(opponent_choice, num_classes=3)
60
61     opponent_prediction_error = actual_opponent_next_choice_one_hot - predicted_opponent_next_choice_probs
62
63     opponent_prediction_state = opponent_prediction_state.at[prev_opponent_choice].add(
64         opponent_learning_rate * opponent_prediction_error
65     )
66
67     opponent_predicted_probabilities_for_next_trial = jax.nn.softmax(opponent_prediction_state[opponent_choice])
68     opponent_prediction_contribution = jnp.roll(opponent_predicted_probabilities_for_next_trial, shift=1) * opponent_inverse_temperature
69
70     choice_logits = (q_values[:, opponent_choice, choice] * inverse_temperature) + alternation_bias + opponent_prediction_contribution
71
72     agent_state = jnp.concatenate((q_values.flatten(), jnp.array([opponent_choice, choice]), opponent_prediction_state.flatten()))
73
74     return choice_logits, agent_state
```

*Listing 8.* Gemini 2.5 Flash SBB Program.

```
1   def gpt_51_agent(
2       params: chex.Array,
3       choice: int,
4       opponent_choice: int,
5       reward: float,
6       agent_state: Optional[chex.Array],
7   ) -> tuple[chex.Array, chex.Array]:
8
9       learning_rate = jax.nn.sigmoid(params[0])
10      reward_sensitivity = jax.nn.sigmoid(params[1])
11      beta = jnp.exp(params[2])
12      initial_q_value = params[3]
13      decay_rate = jax.nn.sigmoid(params[4])
14      opponent_weight = jax.nn.sigmoid(params[5])choice.
15      my_choice_weight = jax.nn.sigmoid(params[6])choice.
16      opponent_prediction_influence = jax.nn.sigmoid(params[7])
17      regret_learning_rate = jax.nn.sigmoid(params[8])
18
19      if agent_state is None:
20          q_table = jnp.full((3, 3, 3), initial_q_value)
21          my_last_choice = -1
22          opponent_last_choice = -1
23          opponent_last_last_choice = -1
24          opponent_pattern_counts = jnp.full((3, 3, 3), 1.0 / 3.0)
25          agent_state = jnp.concatenate([
26              q_table.flatten(),
27              jnp.array([my_last_choice, opponent_last_choice, opponent_last_last_choice]),
28              opponent_pattern_counts.flatten()
29          ])
30
31      q_table_flat_size = 3 * 3 * 3
32      q_table = jnp.reshape(agent_state[:q_table_flat_size], (3, 3, 3))
33      my_last_choice = jnp.int32(agent_state[q_table_flat_size])
34      opponent_last_choice = jnp.int32(agent_state[q_table_flat_size + 1])
35      opponent_last_last_choice = jnp.int32(agent_state[q_table_flat_size + 2])
36      opponent_pattern_counts = jnp.reshape(agent_state[q_table_flat_size + 3:], (3, 3, 3))
37
38      q_table = q_table * (1 - decay_rate)
39
40      def update_q(q_table_to_update, my_prev_choice, opp_prev_choice, r):
41          current_q_value = q_table_to_update[my_prev_choice, opp_prev_choice, choice]
42          effective_alpha = learning_rate * (1.0 + reward_sensitivity * jnp.abs(r))
43          updated_q_value = current_q_value + effective_alpha * (r - current_q_value)
44          return q_table_to_update.at[my_prev_choice, opp_prev_choice, choice].set(updated_q_value)
45
46      q_table = jax.lax.cond(
47          my_last_choice == -1,
48          lambda qt, *_: qt, # If first trial, don't update Q-table
49          update_q,
50          q_table, my_last_choice, opponent_last_choice, reward
51      )
52
53      def update_regret(q_table_to_update, my_prev_choice, opp_prev_choice, current_opponent_choice_for_regret_calc):
54          rewards_if_chosen = jnp.array([
55              [0.0, -0.1, 0.3],
56              [0.3, 0.0, -0.1],
57              [-0.1, 0.3, 0.0]
58          ])
59
60          hypothetical_rewards = rewards_if_chosen[:, current_opponent_choice_for_regret_calc]
61          previous_q_values_for_state = q_table_to_update[my_prev_choice, opp_prev_choice, :]
62
63          regret_updates = regret_learning_rate * (hypothetical_rewards - previous_q_values_for_state)
64          unchosen_mask = (jnp.arange(3) != choice).astype(float)
65          updated_q_values_for_state = previous_q_values_for_state + regret_updates * unchosen_mask
66
67          return q_table_to_update.at[my_prev_choice, opp_prev_choice, :].set(updated_q_values_for_state)
68
69      # Apply regret update if a previous choice exists
70      q_table = jax.lax.cond(
71          my_last_choice == -1,
72          lambda qt, *_: qt, # If first trial, don't update regret
73          update_regret,
74          q_table, my_last_choice, opponent_last_choice, opponent_choice
75      )
76
77      opponent_pattern_counts = opponent_pattern_counts * (1 - decay_rate)
78
79      # Only update if there was a sufficient history of opponent choices (at least two previous choices available).
80      def update_opponent_patterns(counts, last_last_c, last_c, current_c):
81          return counts.at[last_last_c.astype(int), last_c.astype(int), current_c].add(1)
82
83      opponent_pattern_counts = jax.lax.cond(
84          opponent_last_last_choice != -1, # Check if there is a valid sequence (prev_prev -> prev -> current).
85          lambda opc, l_l_c, l_c, c_c: opc.at[l_l_c.astype(int), l_c.astype(int), c_c].add(1),
86          lambda opc, *_: opc,
87          opponent_pattern_counts, opponent_last_last_choice, opponent_last_choice, opponent_choice
88      )
89
90      # --- Decision making for the next trial ---
91      current_my_choice = choice
92      current_opponent_choice = opponent_choice
```

```
93
94    q_values_specific_state = q_table[current_my_choice, current_opponent_choice, :]
95    q_values_opponent_focused = jnp.mean(q_table[:, current_opponent_choice, :], axis=0)
96    q_values_my_focused = jnp.mean(q_table[current_my_choice, :, :], axis=0)
97
98    combined_q_values = (1.0 - opponent_weight - my_choice_weight) * q_values_specific_state + \
99                        opponent_weight * q_values_opponent_focused + \
100                       my_choice_weight * q_values_my_focused
101
102   # --- Opponent prediction component ---
103   current_pattern_slice = opponent_pattern_counts[opponent_last_choice, current_opponent_choice, :]
104   opponent_prediction_probs = current_pattern_slice / (jnp.sum(current_pattern_slice) + 1e-6)
105
106   reward_matrix = jnp.array([
107       [0.0, -0.1, 0.3],
108       [0.3, 0.0, -0.1],
109       [-0.1, 0.3, 0.0]
110   ])
111
112   # Calculate the expected value for each of the agent's actions (0, 1, 2)
113   # with the opponent's predicted probabilities. This is the "fictitious play" component.
114   fictitious_play_expected_values = jnp.dot(reward_matrix, opponent_prediction_probs)
115
116   # Combine all influences
117   final_q_values_for_decision = (1.0 - opponent_prediction_influence) * combined_q_values + \
118                                 opponent_prediction_influence * fictitious_play_expected_values
119
120   choice_logits = final_q_values_for_decision * beta
121
122   # Update agent state for the next trial
123   new_my_last_choice = choice
124   new_opponent_last_choice = opponent_choice
125   new_opponent_last_last_choice = opponent_last_choice # Current opponent_last_choice becomes next opponent_last_last_choice
126   new_agent_state = jnp.concatenate([
127       q_table.flatten(),
128       jnp.array([new_my_last_choice, new_opponent_last_choice, new_opponent_last_last_choice]),
129       opponent_pattern_counts.flatten() # Use the updated opponent_pattern_counts
130   ])
131
132   return choice_logits, new_agent_state
```

*Listing 9.* GPT 5.1 SBB Program.

```python
1  def gpt_oss_120b_irps(
2      params: chex.Array,
3      choice: int,
4      opponent_choice: int,
5      reward: float,
6      agent_state: Optional[chex.Array],
7  ) -> tuple[chex.Array, chex.Array]:
8
9      learning_rate_q, learning_rate_opponent, inverse_temperature, decay_rate, gain_loss_asymmetry_reward, opponent_model_weight,
           persistence_weight, persistence_decay_rate, switch_bonus, reward_sensitivity_to_switch = params
10
11     if agent_state is None:
12        initial_opponent_frequency = 1/3
13        agent_state = jnp.array([0.0, 0.0, 0.0, initial_opponent_frequency, initial_opponent_frequency, initial_opponent_frequency, 0.0,
           0.0, 0.0, -1.0])
14
15     q_values = agent_state[:3]
16     opponent_choice_frequencies = agent_state[3:6]
17     persistence_trace = agent_state[6:9]
18     previous_choice = agent_state[9]
19
20     # Apply gain-loss asymmetry from prospect theory to the reward itself.
21     reward_transformed = jnp.where(reward >= 0, reward, gain_loss_asymmetry_reward * reward)
22
23     q_values = q_values * (1 - decay_rate)
24     prediction_error = reward_transformed - q_values[choice]
25     q_values = q_values.at[choice].set(q_values[choice] + learning_rate_q * prediction_error)
26     # Apply a "regret" update to unchosen actions. This models counterfactual learning.
27     unchosen_mask = 1.0 - jax.nn.one_hot(choice, num_classes=3)
28     q_values = q_values - unchosen_mask * learning_rate_q * prediction_error
29
30     # Update opponent's choice frequencies with a recency bias (exponentially weighted moving average)
31     opponent_choice_one_hot = jax.nn.one_hot(opponent_choice, num_classes=3)
32     opponent_choice_frequencies = (1 - learning_rate_opponent) * opponent_choice_frequencies + learning_rate_opponent *
           opponent_choice_one_hot
33     # Normalize opponent choice frequencies to ensure they sum to 1, representing a probability distribution.
34     opponent_choice_frequencies /= jnp.sum(opponent_choice_frequencies)
35
36     # Calculate expected rewards for each of agent's possible choices based on opponent's predicted strategy
37     reward_matrix = jnp.array([
38         [0.0, -0.1, 0.3],
39         [0.3, 0.0, -0.1],
40         [-0.1, 0.3, 0.0],
41     ])
42     expected_rewards_from_opponent_model = jnp.dot(reward_matrix, opponent_choice_frequencies)
43
44     # Combine Q-values and expected rewards from opponent model, weighted by opponent_model_weight
45     combined_values = q_values + opponent_model_weight * expected_rewards_from_opponent_model
46
47     # Update the persistence trace. All traces decay, and the chosen action's trace is incremented.
48     persistence_trace = persistence_trace * (1 - persistence_decay_rate)
49     persistence_trace = persistence_trace.at[choice].set(persistence_trace[choice] + 1.0) # Increment trace for chosen action
50     # Add the persistence trace to the combined values, acting as a direct action bias, scaled by the persistence_weight.
51     combined_values += persistence_weight * persistence_trace
52
53     # Add a switching bonus: if the current action is different from the previous one, add a bonus.
54     is_switch = jax.nn.one_hot(choice, num_classes=3) != jax.nn.one_hot(jnp.array(previous_choice, dtype=jnp.int32), num_classes=3)
55     modulated_switch_bonus = switch_bonus + reward_sensitivity_to_switch * jnp.abs(prediction_error)
56     switch_effect = jnp.where(previous_choice == -1, jnp.zeros(3), is_switch * modulated_switch_bonus)
57     combined_values += switch_effect
58
59     choice_logits = inverse_temperature * combined_values
60
61     # Update the full agent state
62     agent_state = jnp.concatenate((q_values, opponent_choice_frequencies, persistence_trace, jnp.array([choice])))
63
64     return choice_logits, agent_state
```

*Listing 10.* GPT OSS 120B SBB Program.

