# OpenReview forum: "Discovering Differences in Strategic Behavior between Humans and LLMs"
_ICML.cc/2026/Conference — ICML 2026 regular_

### Official Review · Reviewer_kWC6 · 2026-02-15

**Soundness:** 3
**Presentation:** 3
**Significance:** 3
**Originality:** 3
**Overall Recommendation:** 5
**Confidence:** 4

**Summary:**

The authors explore whether LLMs  have different strategic behavior compared to humans, and whether behavior game theory can capture LLM style interactions. To do this, they examine iterative rock paper scissors (IRPS) with data from human performance (from prior work) and LLM interactions (using different closed sourced models). With this behavior data, they use a program discovery tool (AlphaEvolve) to generate reasonably high fidelity programmatic representations of the observed strategies, which can be used to interpret and understand said data. They fit the various LLM data as well as human data to generate programs which they then analyze for their similarites and differences.

They find that all program representations include a value based learning component and an openent modeling. Human + most LLMs (besides GPT-OSS-120b) use a 3 dimensional Q-value learning matrix (OSS uses 1 dimenaionl Q-value). For opnentn modeling, humans and GPT-OSS-120b use single dimensional openent trackingi (overall frequency of opnenents move) while the other LLMs use more complex (conditional based on prior moves or 3x3x3) - suggesting that these more complex models perform better because they have a more complex modeling of the users.

**Compliance With Llm Reviewing Policy:**

Affirmed.

**Key Questions For Authors:**

1) How do you verify AlphaEvolve is giving you reasonably predictive programs of the underlying behavior?
2) Do you expect program differences between different humans? The claim that humans vs LLM behave differently rest upon looking (I think ) an aggregate of human produced data vs a single LLM. If you took, say a professional Rock Paper Scissor player (which do exists) would you expect them to behave in a similar manner to the LLMs?
3) I'm still not fully clear  on Figure 5 - if you could explain the important takeaways that would be helpful.

**Limitations:**

For the limitations it would also be good to note that in this game, the payoffs are well known to both agents - in a lot of other situations the exact payoffs matrix may be unknown.

It would also be nice to note what types of programs/systems AlphaEvolve might have challenges with and what type of strategic games are more amenable to AlphaEvolve vs not.

**Strengths And Weaknesses:**

Strengths:
Soundness: The experiments are mostly sound -> testing the performance across different bots over repeated iterations is sensible and the program analysis into programmatic components is well done.

Presentation:
The presentation is mostly well structured and the motivation is clear. The connection to past works are clear - especially the motivation and use case of AlphaFold.
The interpretability of the program analysis and choosing of the simplest model made sense and connects proper..

Significance: Representing humans vs LLM strategic behavior is an important area of research; as we expect these LLMs/agents to behave in other settings, understanding how they may strategically reason with other humans/bots/agents is an important area of work.

Originality: The use of program analysis to understand game theoretical context was new to me - though there may be other works doing similar things or end up using Behavioral Game Theory adequately well.


Weaknesses:

Some descriptions of the experiments are not super clear.

1) The different easy/medium/hard bots aren't described in the main body. They are mentioned in Appendix C but no direct reference to that is given, hence it was hard to understand what these are doing. Moreover, the easy/medium/hard designation just seems to be based on the types of feedback it takes in? Is that a sufficient justification for the difficulty facing a bot (the reason there's a slight distinction is that in many situations (think repeated prisoner's dilemma, strategies like tit-for-tat which are 'less' complex still perform quite well).

2) The distinction between adaptive and non-adaptive wasn't completely clear -> based on the description in the text, the main difference are restrictions on the history lookback?

3) Figure 2: The human baseline is mentioned in the text but the figures only have nash/oracle? Is there a line for strategic human performance? It's not clear which one was CS-EWA vs RNN baselines are in the Figures as well

4) For Figure 3, I'm having a hard time understanding what "the relative improvement in twofold cross-validated normalized likelihood over the Nash equilibrium baseline" is supposed to mean and how I should be thinking about it.

5) Figure 5 is also not very clear to me: the color scaling label is confusing (the same-ish number having different colors right next to each other). It's hard for me to understand the takeaway - is it supposed to be that models with similar underlying strategies are more able to be predicted by each other.

6) The soundness relies very heavily on AlphaEvolve generating good representations of the program, which may not always hold in practice. Commenting on that would be helpful.

---

> ### Author Rebuttal · Authors · 2026-03-31
>
> We thank the reviewer for the careful reading and detailed feedback. We address each point below.
>
> ### Clarification on Bots
>
> Nonadaptive bots employ fixed transition-based strategies that map a small amount of historical context to predetermined next moves, without updating based on opponent patterns. Adaptive bots maintain an internal lookup table tracking opponent choice counts conditioned on history, using it to predict and counter future moves. The key distinction is that adaptive bots are designed to exploit their opponent, while nonadaptive bots follow fixed statistical rules regardless of opponent behavior.
>
> Easy/medium/hard designations are a subjective categorization intended to help the reader interpret results. Within each bot class, the complexity ranking reflects the history window size used by the bot's internal lookup table: a bot conditioning on more history is harder to exploit because it requires the opponent to track longer dependencies. As we note at Appendix L1090-1096, complexity rankings are only meaningful within a class—one cannot directly compare a hard nonadaptive bot to an easy adaptive bot, as they differ fundamentally in exploitability structure. We are happy to add this explanation to the main paper.
>
> ### Clarifying Figure 2
>
> Section 4.1 and Figure 2 analyze only raw win rates of each agent against each bot; no behavioral modeling has occurred at this point. AlphaEvolve, CS-EWA, and the RNN baseline do not appear until Figure 3. Human strategic performance is displayed in the leftmost column of Figure 2, with mean win rates and 95% confidence intervals shown across bots of increasing complexity (top row) and over time within a game (bottom row). The Nash/random and oracle lines serve as lower and upper performance benchmarks.
>
> ### Clarifying Figure 3
>
> Figure 3 reports fit quality of each behavioral modeling method (AlphaEvolve, RNN, CS-EWA) as improvement in normalized likelihood over the Nash equilibrium baseline. The Nash equilibrium predicts each action with probability 1/3, achieving a normalized likelihood of 1/3 on all datasets. A model that better captures actual choices will assign higher probability to the chosen action on average, yielding a higher normalized likelihood. Each point represents a single held-out test game, allowing the reader to see both mean improvement and variability across games.
>
> ### Soundness of AlphaEvolve Programs (Q1)
>
> We are unsure what the reviewer means by a "good representation" of a program, and would welcome clarification.
>
> We validate AlphaEvolve's programs along two dimensions. First, predictive quality: all programs are evaluated on a held-out test set via two-fold cross-validation; Figure 3 shows AlphaEvolve matches or exceeds a highly flexible RNN baseline while substantially outperforming the BGT-inspired CS-EWA baseline. Second, interpretability is assessed via Halstead effort, which measures the cognitive effort required to understand a program. The SBB programs are provided in Appendix F for direct inspection.
>
> ### Individual Differences Between Humans (Q2)
>
> Our approach fits a single program to the aggregate human population rather than per individual, since most participants complete only a single game, making per-subject fitting unreliable. The discovered human SBB program therefore characterizes average human strategic behavior. It is plausible that expert players would exhibit more sophisticated opponent modeling, potentially resembling frontier LLM programs, and we will note this in the limitations section.
>
> ### Clarifying Figure 5 (Q3)
>
> Figure 5 is a cross-generalization matrix: each column represents an SBB program discovered for a given agent, and each row represents the dataset on which that program is evaluated with parameters refit.
>
> The key takeaways are twofold. First, diagonal entries dominate within each row, confirming each program best predicts the agent it was optimized for. Second, high off-diagonal scores among Gemini 2.5 Flash, Gemini 2.5 Pro, and GPT 5.1 indicate behavioral similarity across frontier LLMs, while a significant symmetric gap between all LLM programs and the human program confirms strategic differences.
>
> We will revise the color scaling so each row uses its own scale, making within-row comparisons clearer.
>
> ### Additional Limitations
>
> We will incorporate both suggestions.
>
> *On known payoffs:* neither the LLMs playing IRPS nor AlphaEvolve were informed of specific payoffs, and this paper uses non-standard payoffs. However, frontier LLMs are likely aware of the general IRPS payoff structure and may not display the same strategic abilities in fully novel games.
>
> *On scope:* AlphaEvolve is most naturally suited to repeated interaction settings with well-defined action spaces and sufficient behavioral data; games with large or continuous state/action spaces, complex dynamics, or sparse data would pose challenges.

---

> > ### Author Rebuttal · Reviewer_kWC6 · 2026-04-03
> >
> > Thank you for your detailed response. I have no further questions.

---

### Official Review · Reviewer_mwS2 · 2026-02-28

**Soundness:** 2
**Presentation:** 4
**Significance:** 2
**Originality:** 3
**Overall Recommendation:** 4
**Confidence:** 4

**Summary:**

LLMs are increasingly used to directly interact with humans, and as proxies to study human behaviors in silico. This trend motivates this paper's main investigation, which is to study how human strategic behavior differs from large language models' by analyzing how humans and LLMs play iterated Rock Paper Scissors games. Existing behavior game theory models are mostly based on human cognitive assumptions. They may fail to capture patterns specific to LLMs. So the authors use AlphaEvolve to learn interpretable models directly from gameplay data. The authors find humans and LLMs both use value-based learning and opponent modeling. And compared with humans, best-performing LLMs create higher-dimensional models of their opponents. This allows them to detect and make use of patterns faster than humans. So as a result, advanced models achieve higher win rates against structured bots, whereas smaller models struggle. The main takeaway include: advanced models are not faithful proxies of human strategical behavior, and introduced automatic symbolic model discovery as a method to study humans/LLMs behavior.

**Compliance With Llm Reviewing Policy:**

Affirmed.

**Final Justification:**

The paper studies differences between human and LLM strategic behavior through iterated RPS games and introduces alpha evlove as a tool for behavioral analysis. I find the experimental setup careful and the empirical comparisons credible, with strong baselines and a thoughtful treatment. The use of alpha evolve to extract interpretable behavioral programs is a key strength and represents a meaningful methodological contribution that could extend beyond this specific setting. While the originality lies more in the application and framing than in the underlying method, the interpretability and cross-disciplinary relevance increase its impact. The presentation is generally clear, though the methods section could benefit from a more accessible high level overview and a concrete example to guide the reader.

My main concern was about, and is still about generalization beyond IRPS. The rebuttal clarified the intended scope and committed to revising the framing to avoid overgeneralization. It also provided a reasonable justification for focusing on a single game due to data requirements, and included an initial robustness check on payoff perturbations. While I still strongly believe additional games would strengthen the claims, i also understand the difficulty of finding a large-scale dataset of other games in similar format, and i am satisfied that the authors have appropriately scoped their contributions and articulated a clear research trajectory. Given the solid experimental evidence within IRPS and the value of the proposed methodology, I update my evaluation to weak accept.

**Key Questions For Authors:**

My concerns are mostly around generalization. I appreciate the authors have acknowledged this in the limitation and future work section, but I do feel at least one more game is needed for this broad claim of Strategic Behavior. See Strengths And Weaknesses. If extra experiment is done during rebuttal and the main conclusions still hold, I would raise my score.

**Limitations:**

yes

**Strengths And Weaknesses:**

- Soundness: I think the experiments are mostly careful and well done. The human dataset is quite large, and the LLM setup closely matches human setup. Comparisons to strong baselines add credibility. I also appreciate the paper's explicit tradeoff between fit and simplicity. That said, I’m concerned that all conclusions come from a single game with fixed bots. I don’t see robustness checks across different payoff structures or opponent types. See Significance for similar comments.
- Presentation: I think the paper is well-written in general. I like the introduction as it motivate the paper clearly, and I resonate with the motivation personally. The use of a pareto frontier and the simplest-but-best rule also makes sense. But I do find myself slow down a bit in the method section. Largely because the AlphaEvolve setup and bilevel optimization are dense. I think a short, high-level summary of the full pipeline and a concrete overview of one discovered program would make it easier to digest and contextualize.
- Significance: I believe the question is important. If LLMs are used as social agents, understanding structural differences matters; and delegation to LLM agents is without doubt a growing trend. The finding that frontier LLMs maintain more complex opponent models than human is interesting and meaningful, especially given the speed of LLM advancement. But I’m not convinced that the result can generalize beyond IRPS. Everything in this work is built around IRPS, so I cannot be sure how much this says about broader strategic reasoning. Strategic reasoning is a bid word, and there are much more than IRPS.
- Originality: I really like applying automated symbolic program discovery to behavioral game theory. This has implications beyond ML community. This interpretability focus is a strength. However, I think the main originality as the application and framing, not the method itself. AlphaEvolve is existing work, and the modeling framework is largely inherited.

Some actionable suggestions: I would strongly encourage the authors adding at least one more repeated game with a different underlying structure. Rich work can be found from economic games, such as Prisoner Dillema/Public Goods Game etc; or a maybe some coordination games, to test whether the same structural differences appear. I would also like to see some robustness checks: vary payoff matrices, bot strategies, and report whether opponent-model dimensionality remains stable. This ablation study should not be too heavy, yet it would help to analyze stability.
Some possible referense:
- Piatti, Giorgio, et al. "Cooperate or collapse: Emergence of sustainable cooperation in a society of llm agents." Advances in Neural Information Processing Systems 37 (2024): 111715-111759.
- Li, Yuxuan, and Hirokazu Shirado. "Spontaneous giving and calculated greed in language models." Proceedings of the 2025 Conference on Empirical Methods in Natural Language Processing. 2025.
- Sreedhar, Karthik, et al. "Simulating cooperative prosocial behavior with multi-agent llms: Evidence and mechanisms for ai agents to inform policy decisions." Proceedings of the 30th International Conference on Intelligent User Interfaces. 2025.

---

> ### Author Rebuttal · Authors · 2026-03-31
>
> We thank the reviewer for the careful reading and constructive feedback. We particularly appreciated the positive assessment of the experimental rigor and originality. We address each point below.
>
> ### Generalization/Significance of Results Beyond IRPS
>
> We acknowledge that the paper's framing invited the interpretation that we make broad claims about strategic reasoning beyond IRPS. In response, we will modify the title to explicitly reference IRPS and introduce it directly in the key research question at L25. Within that context, we address the concern in three parts.
>
> **On the significance of findings within IRPS**
>
> We argue that our findings are interesting in their own right.
>
> - We exhibit a concrete scenario where frontier LLMs display deeper strategic behavior than average humans, directly contrasting [1], who studied a similar IRPS setting and found that even GPT-4 remained worse than humans. This reversal highlights rapid frontier LLM advancement and has direct implications for AI safety.
> - This concretely demonstrates that frontier LLMs cannot serve as faithful digital twins for humans in strategic settings, a practically important conclusion for social science researchers who rely on LLM proxies.
> - We show that AlphaEvolve, previously demonstrated for cognitive modeling in neuroscience [2], generalizes to modeling strategic interactions, a methodological contribution largely independent of the specific game.
>
> **On extending to additional games:**
>
> Our bilevel evolutionary search via AlphaEvolve requires large behavioral datasets to produce generalizable programs. Our human IRPS dataset contains 129,087 choices and each LLM dataset contains 90,000 choices. For comparison, Castro et al. 2024 [2] relied on datasets of 617,871 (human), 1,087,140 (rat), and 68,000 (fly) choices. Repeated matrix game datasets of comparable scale are rare. We evaluated several candidate datasets but found none that met our requirements of scale and repeated interaction structure. Collecting such data for a second game is a substantial undertaking beyond the scope of this paper.
>
> In light of the above, this paper prioritizes a careful, deep analysis of IRPS, over a shallower analysis across multiple games:
>
> - We consider both humans and 4 LLMs
> - We analyze the win rates of all agents.
> - We validate AlphaEvolve against multiple strong baselines.
> - We validate that synthetic win rate statistics of the discovered models match ground-truth win rates.
> - We examine cross-generalization of discovered models.
> - We verify the prevalence of key mechanisms across the Pareto frontier.
> - We provide detailed semantic analysis of discovered programs.
>
> We thank the reviewer for the references and will cite/discuss them within the related work.
>
> **On the broader research trajectory:**
>
> Behavioral game theory has progressed from experts manually crafting models for single games [3, 4], to ML methods for single-shot games [5], to recent work combining ML with existing theory to learn models generalizing across games [6]. Building truly general behavioral models of human and LLM strategic behavior would require a large-scale dataset spanning a diverse set of repeated games—an exciting future direction. Our paper advances this agenda by (1) introducing automated symbolic model discovery to behavioral game theory, and (2) providing the first structural comparison of frontier LLM and human behavior in a repeated strategic setting.
>
> ### Robustness of SBB Programs to Variations in Payoff Matrix
>
> We assess the robustness of the SBB program for Gemini 2.5 Flash to variations in payoff matrices by sampling games from an IRPS variant where payoffs were multiplied by 10. The SBB program achieves a likelihood of **0.462** under this variant. Note that the SBB program likelihood on the original dataset was **0.491**. This suggests the discovered behavioral models are robust to payoff perturbations.
>
> ## Improvements to Presentation of Method
>
> To improve readability of the methods section, if the paper is accepted, we will use the extra page to add a figure illustrating the AlphaEvolve modeling framework, together with an example program. The caption will provide a short, high-level summary.
>
> [1] Fan et al. Can LLMs Serve as Rational Players in Game Theory? AAAI 2024.
>
> [2] Castro et al. Discovering Symbolic Cognitive Models from Human and Animal Behavior. ICML 2025.
>
> [3] McKelvey et al. Quantal Response Equilibria for Normal Form Games. Games and Economic Behavior 1995.
>
> [4] Stahl et al. On Players′ Models of Other Players: Theory and Experimental Evidence. Games and Economic Behavior 1995.
>
> [5] Hartford et al. Deep Learning for Predicting Human Strategic Behavior. NeurIPS 2016.
>
> [6] Zhu et al. Capturing the complexity of human strategic decision-making with machine learning. Nature Human Behaviour 2025.

---

> > ### Author Rebuttal · Reviewer_mwS2 · 2026-04-03
> >
> > Thank you for the detailed response. I will raise my score. With that said, I still suggest the author spend a bit more effort in the writing to better position IRPS (e.g. how it can be a good representative of a larger problem space), which will make readers like me give more credits to the paper's generalizability.

---

### Official Review · Reviewer_vS9a · 2026-03-12

**Soundness:** 4
**Presentation:** 4
**Significance:** 4
**Originality:** 4
**Overall Recommendation:** 5
**Confidence:** 4

**Summary:**

This paper studies how large language models (LLMs) behave strategically in repeated interactions compared with humans. The authors adopt the iterated rock–paper–scissors game as the studied scenarios, and analyze gameplay data from human participants and several frontier LLMs. They then apply AlphaEvolve, an automated program discovery framework, to infer interpretable behavioral programs.

**Compliance With Llm Reviewing Policy:**

Affirmed.

**Final Justification:**

Insightful work.

**Key Questions For Authors:**

- The current study focuses on the iterated rock–paper–scissors setting. It would be interesting to know whether the authors have considered applying AlphaEvolve to other behavioral game environments (e.g., trust games, coordination games, or public goods games). Do similar strategic patterns emerge in those settings as well, or does the structure of the discovered programs depend strongly on the specific game? Exploring this direction could help clarify the generality of the proposed approach.

- The paper interprets the discovered programs in terms of mechanisms such as Q-value learning and opponent modeling. I am curious how the authors think about the behavioral interpretation of these programs. To what extent can these mechanisms be meaningfully compared with human decision processes? For example, do the authors view similarities such as Q-value–like updates as reflecting shared explanatory structures with human strategies, or primarily as useful abstractions for describing observed behavior?

**Limitations:**

Yes

**Strengths And Weaknesses:**

**Strengths:** Very Impressive Paper!
- Important and timely question. The paper studies whether LLMs exhibit human-like strategic behavior in repeated interactions, which is relevant for both behavioral science and the growing use of LLM agents as human proxies.
- Interesting methodology. The use of AlphaEvolve to automatically discover interpretable behavioral programs is novel and goes beyond standard fixed behavioral models.
- Insightful empirical analysis. The results show systematic differences between human and LLM strategies in repeated rock–paper–scissors, and the paper attempts to interpret these differences through learned mechanisms such as value learning and opponent modeling.

**Weaknesses:**
- Limited scope of the experimental setting. All conclusions are drawn from a single stylized game (IRPS), so it is unclear whether the findings generalize to richer strategic environments.
- The choice of interpretability metric could be further discussed. While code complexity measures like Halstead effort capture the syntactic verbosity of the generated programs, they may not faithfully reflect the underlying behavioral or strategic complexity of the agents. A highly concise block of code could still encode a cognitively demanding strategy, and vice versa. The authors might consider exploring metrics that measure the intrinsic complexity of the decision logic itself, for instance, Santos et al. (Nature 2018, https://doi.org/10.1038/nature25763).

---

> ### Author Rebuttal · Authors · 2026-03-31
>
> We are grateful for the reviewer's careful reading and highly positive assessment.
>
> ### Generalization Beyond IRPS
> > It would be interesting to know whether the authors have considered applying AlphaEvolve to other behavioral game environments.
>
> We have considered this carefully. Our bilevel evolutionary search via AlphaEvolve requires large behavioral datasets to discover generalizable programs; repeated matrix game datasets matching the scale of our IRPS data (129,087 human choices) are rare, and we found no suitable candidate for a second game. Given this constraint, we prioritized depth of analysis over breadth: we consider both humans and 4 LLMs, rigorously validate AlphaEvolve against multiple strong baselines, examine cross-generalization across all agent pairs, verify the prevalence of key mechanisms across the full Pareto frontier, and provide detailed semantic analysis grounded in BGT concepts.
>
> We refer the reviewer to our responses to reviewer **mwS2** for a fuller discussion of the significance of our IRPS findings and the broader research trajectory.
>
> ### Choice of Halstead Metric for Interpretability
> > Halstead effort captures syntactic verbosity... a highly concise block of code could still encode a cognitively demanding strategy, and vice versa.
>
> We fully agree that Halstead effort is a heuristic, and we adopted it following [1]. We agree it would be interesting to explore metrics more directly reflecting cognitive demand, such as the Boolean-complexity-based metric introduced by [2] and applied by [3]. However, applying such a metric to our setting raises non-trivial challenges. The Boolean complexity framework was developed for binary input data that naturally can be described by concise Boolean formulas. While a translation of Python programs into Boolean formulas is theoretically possible by leveraging machine-level binary representations, the resulting formulas would likely be much larger than those considered by [2] (up to 12 literals) or [3] (16 literals).
>
> Furthermore, it remains unclear whether LLMs share the same notion of cognitive demand as humans. Recent work by [4] demonstrates that frontier LLMs struggle with tasks that are trivial for humans, such as character counting, first-order logic evaluation, and travel planning, despite excelling at problems humans find difficult. This suggests that human-centric complexity metrics do not straightforwardly transfer to LLM behavioral programs. We will add this discussion to the limitations section and note that developing better-suited interpretability metrics for both human and LLM behavioral programs is an interesting direction for future work.
>
> ### Behavioral Interpretation of Discovered Programs
> > To what extent can these mechanisms be meaningfully compared with human decision processes? Do the authors view similarities such as Q-value-like updates as reflecting shared explanatory structures with human strategies, or primarily as useful abstractions for describing observed behavior?
>
> We view the discovered mechanisms primarily as useful abstractions: structured, testable hypotheses validated by predictive fit, rather than ground-truth accounts of underlying decision processes. As we discuss in the paper (L290-302), the programs are best understood in the spirit of scientific modeling; they explain and predict behavior without necessarily implying that the agent internally implements the described computation.
>
> That said, the Q-value-like updates do share explanatory structure with human strategies in a functional sense, as both describe the same computation of tracking and updating action values based on received rewards, producing similar behavioral predictions. Whether this reflects a shared underlying process or a convergent functional solution arrived at by very different means is an open and interesting question that mechanistic interpretability techniques could, in principle, help address.
>
> [1] Castro et al. Discovering Symbolic Cognitive Models from Human and Animal Behavior. ICML 2025.
>
> [2] Feldman. Minimization of Boolean complexity in human concept learning. Nature 2000.
>
> [3] Santos et al. Social norm complexity and past reputations in the evolution of cooperation. Nature 2018.
>
> [4] Malek et al. Frontier LLMs Still Struggle with Simple Reasoning Tasks. arXiv 2025.

---

> > ### Author Rebuttal · Reviewer_vS9a · 2026-04-02
> >
> > Thank you for these insightful discussions. I would be glad to see them reflected in the final version. Several interesting questions could also be explored in the future directions suggested by this work.

---

### Official Review · Reviewer_a3ap · 2026-03-12

**Soundness:** 1
**Presentation:** 2
**Significance:** 2
**Originality:** 3
**Overall Recommendation:** 4
**Confidence:** 3

**Summary:**

This paper aims at understanding structural differences in strategic behaviour between humans and Large Language Models (LLMs). The core objective is achieved by applying AlphaEvolve, a program discovery tool, to generate interpretable programmatic models from behavioural data in the Iterated Rock-Paper-Scissors (IRPS) game.

The empirical study evaluates humans and multiple LLMs playing against a panel of 15 bots of varying complexity. AlphaEvolve is used to search a space of programmatic behavioural models, optimising for both predictive likelihood and code simplicity, to capture and compare the underlying strategies of each agent type.

**Compliance With Llm Reviewing Policy:**

Affirmed.

**Final Justification:**

After checking author rebuttal and other reviews, I decided to raise my score.

**Key Questions For Authors:**

1. To what extent are the discovered structural features (e.g., the 3x3 transition matrix modelling) sensitive to variations in prompt framing, persona adoption, or the representation of game history?
2. Are there any auxiliary results (e.g., from internal state probing or logit analysis) suggesting that the LLMs are computationally performing operations analogous to the discovered programmatic logic? Clarifying this would help in situating the work's contribution to mechanistic interpretability.

**Limitations:**

Yes

**Strengths And Weaknesses:**

**Strengths**
1. The methodology of using AlphaEvolve to synthesise human-readable programs as behavioural models seems original.
2. This submission addresses whether LLMs serve as accurate proxies for human strategic behaviour in interactive settings. This topic is central to AI safety and alignment.

**Weaknesses**
1. The empirical scope is constrained to a single, relatively simplistic game (Iterated Rock-Paper-Scissors). Extrapolating broad claims about LLMs exhibiting "deeper strategic behavior than humans" from this narrow setting is insufficiently justified, as it remains unclear if these findings generalize to other types of activities, e.g., cooperative games or environments with continuous action spaces.
2. The methodology assumes that the high-likelihood programmatic models discovered by AlphaEvolve indicate the causal mechanistic drivers of behaviours. While these programs offer descriptive value, there is a gap in proving that these post-hoc symbolic approximations accurately map to the internal computational processes of the evaluated LLMs.

---

> ### Author Rebuttal · Authors · 2026-03-31
>
> We thank the reviewer for recognizing the importance of the topic and the originality of our approach.
>
> ### Extrapolating Beyond IRPS
> Our paper does not extrapolate claims about LLM strategic behavior in IRPS to general LLM strategic behavior. Our claims about deeper/more sophisticated LLM strategic behavior (abstract L25; introduction final bullet; discussion L424) always explicitly reference IRPS. We are happy to add further qualification to any specific claims the reviewer finds overly broad.
>
> Nevertheless, we acknowledge that the paper's framing invited the interpretation that we make broad claims about strategic reasoning beyond IRPS. In response, we will modify the title to explicitly reference IRPS and introduce it directly in the key research question at L25.
>
> For a complete discussion of the broader significance of our IRPS results, extensions to other games, and contextualization within the broader research trajectory of behavioral game theory, please see our response to **Reviewer mwS2**.
>
> **IRPS as a Testbed.**  IRPS is an actively used benchmark across cognitive science, neuroscience, and AI precisely because it isolates opponent modeling and adaptive sequential learning in a setting with well-understood game-theoretic properties. Many well-regarded papers study human and agent behavior exclusively within IRPS [3–7]. Like these works, our paper emphasizes depth of analysis over breadth.
>
> **Programs as Mechanistic Hypotheses.** The paper explicitly does not assume that the programs are ground truth mechanistic models of how agents make decisions in IRPS; see  L290-302 (Section 4.3). Rather, we view them as structured models that are testable, mechanistic hypotheses of behavior.
>
>
> Our analysis methods are designed accordingly: Section 4.3.2 treats SBB programs as mathematical models of behavior, examining the ability of each to predict behavior of other agents, an approach equally applicable to neural network models. Section 4.3.3 validates that opponent modeling and value-based learning are not artifacts of the SBB programs alone, by checking their prevalence across the full Pareto frontier (Appendix Fig. 7).
>
> We agree that closing the gap between post-hoc symbolic approximations and internal computational processes would require mechanistic interpretability techniques such as internal state probing or logit analysis. This is outside the scope of the current paper, and infeasible given that the frontier LLMs examined (Gemini 2.5 Flash/Pro, GPT 5.1) are proprietary. We release the SBB programs in Appendix F to enable researchers with model access to build on this work. Nevertheless, we agree that this paper’s methods/results are potentially interesting from the perspective of mechanistic interpretability. We are happy to add this discussion to the limitations and future work section.
>
> Finally, we note that predictive modeling is a valid approach to understanding behavior used broadly in neuroscience and behavioral game theory, even absent mechanistic causal explanations.
>
> **Sensitivity Analysis**
> Unfortunately, rerunning the full analysis pipeline within the rebuttal period is not feasible, but we provide the following context.
> - *Prompt design*: The prompt is designed to match the instructions provided to the human participants in the original [3] dataset as closely as possible, removing only references to specific details of the human data collection process such as user interface pointers.
> - *Personas*: We are interested in characterizing the emergent strategic behavior of LLMs, and therefore do not explore persona-based prompting. Prior work has found that LLM strategic behavior can be sensitive to persona-based prompting with effects varying across LLMs [8]; we expect our results to be similarly sensitive.
> - *Game history representation*: Due to computation constraints, we did not explore the impact of the game representation. Our representation of the game is in line with [9, 10].
>
>
> [1] Fan et al. Can LLMs Serve as Rational Players in Game Theory? AAAI 2024.
>
> [2] Castro et al. Discovering Symbolic Cognitive Models from Human and Animal Behavior. ICML 2025.
>
> [3] Brockbank & Vul. Repeated RPS play reveals limits in adaptive sequential behavior. Cognitive Psychology 2024.
>
> [4] Dyson et al. Negative outcomes evoke cyclic irrational decisions in RPS. Scientific Reports 2016.
>
> [5] Moerel et al. Neural decoding of competitive decision-making in RPS. SCAN 2025.
>
> [6] Lanctot et al. Population-based Evaluation in Repeated RPS as a Benchmark for Multiagent RL. TMLR 2023.
>
> [7] Cross et al. Understanding Human Limits in Pattern Recognition: Sequential Reasoning in RPS. CCN 2025.
>
> [8] Lorè & Heydari. Strategic behavior of LLMs and the role of game structure vs. contextual framing. Scientific Reports 2024.
>
> [9] Qian et al. Strategic Tradeoffs Between Humans and AI in Multi-Agent Bargaining. arXiv 2025.
>
> [10] Akata et al. Playing repeated games with large language models. Nature Human Behaviour 2025.

---

> > ### Author Rebuttal · Reviewer_a3ap · 2026-04-04
> >
> > Thank you to the authors for their reply. I’ve also checked carefully their response to reviewer mw52. I will raise my score.

---

> > > ### Author Response · Authors · 2026-04-08
> > >
> > > We sincerely thank the reviewer for their constructive feedback and for carefully reading our response. We are glad we were able to address your concerns and will incorporate the results and discussion above to improve the paper. We also note that the Final Acknowledgement form is now available via "Edit Review", in case that is useful.

---

### Decision · Program_Chairs · 2026-04-30

**Decision:**

Accept (regular)

**Comment:**

This paper employs AlphaEvolve to automatically discover interpretable programmatic models of strategic behavior from data in Iterated Rock-Paper-Scissors (IRPS). It compares aggregate human behavior with that of several frontier LLMs, revealing that advanced LLMs exhibit deeper opponent modeling and superior performance against structured bots. The four reviewers unanimously praised the timeliness of the research question (human vs. LLM strategic differences) and the novelty of the methodology—applying automated program discovery to behavioral game theory. They highlighted the careful experimental design, strong baselines, insightful mechanistic analysis (value-based learning and opponent modeling), and the paper’s clear contribution to AI safety, alignment, and computational social science. The weaknesses are minor, and I recommend acceptance.